# Spatially resolved cell atlas of the teleost telencephalon and deep homology of the vertebrate forebrain
Brianna E. Hegarty[1,2,6], George W. Gruenhagen [1,2,6] ✉, Zachary V. Johnson[1,2,3,4,6], Cristina M. Baker [1,2,5] & Jeffrey T. Streelman [1,2] ✉

The telencephalon has undergone remarkable diversification and expansion throughout vertebrate evolution, exhibiting striking variations in structural and functional complexity. Nevertheless, fundamental features are shared across vertebrate taxa, such as the presence of distinct regions including the pallium, subpallium, and olfactory structures. Teleost fishes have a uniquely "everted" telencephalon, which has confounded comparisons of their brain regions to other vertebrates. Here we combine spatial transcriptomics and single nucleus RNA-sequencing to generate a spatially-resolved transcriptional atlas of the *Mchenga conophorus* cichlid fish telencephalon. We then compare cell-types and anatomical regions in the cichlid telencephalon with those in amphibians, reptiles, birds, and mammals. We uncover striking transcriptional similarities between cell-types in the fish telencephalon and subpallial, hippocampal, and cortical cell-types in tetrapods, and find support for partial eversion of the teleost telencephalon. Ultimately, our work lends new insights into the organization and evolution of conserved cell-types and regions in the vertebrate forebrain.

The forebrain houses regions that regulate complex functions such as learning, memory, and social behavior. Despite its conserved functions, this structure has grown larger and increasingly complex over evolutionary time, exhibiting marked variations across vertebrate taxa including the mammalian six-layered neocortex, the dorsal ventricular ridge (DVR) in sauropsids (reptiles and birds), and the "everted" teleost pallium of ray-finned fish (Actinopterygii). As a result, the evolutionary relationships of specific forebrain regions across species are still actively debated.

Several subregions of the teleost telencephalon have been compared to mammalian brain regions involved in regulating social behaviors, including the hippocampus, striatum, and septum[1]. However, establishing homologues between telencephalic subdivisions in fish to those of distantly-related vertebrate clades has been historically challenging, due in part to the "everted" morphology of the teleost telencephalon. The unique outward folding of the teleost pallium during development, known as eversion, leads to an altered arrangement of pallial zones compared to the "evaginated" brains of other vertebrate lineages[2,3]. Multiple proposed models of eversion attempt to identify the counterparts of other vertebrate pallial territories in fish[2–6]. However, these models are unresolved, leaving the evolutionary identities of most divisions of the teleost pallium in question. Modern single cell and spatial omics technologies are powerful tools to address these long-standing questions. Recent studies in different tetrapod lineages have begun to utilize these techniques to investigate vertebrate brain evolution at an unprecedented resolution[7–12]. Advances in methods for comparative analysis of these data have further increased the ability to establish homologous cell-types across distant phyla[13,14].

We recently employed single nucleus RNA-sequencing (snRNA-seq) to link genomic signatures of behavioral evolution to specific cell populations in the cichlid telencephalon[15]. Here, we perform spatial transcriptomics (ST) and map cell populations identified by snRNA-seq to create a spatially resolved transcriptional atlas of the cichlid telencephalon. We then survey the cellular architecture of this brain structure and compare its component cell-types and brain regions across all five major vertebrate lineages. Our work generates new and unanticipated insights into teleost neurodevelopment and vertebrate forebrain evolution.

[1]School of Biological Sciences, Georgia Institute of Technology, Atlanta, GA 30332, USA. [2]Institute of Bioengineering and Bioscience, Georgia Institute of Technology, Atlanta, GA 30332, USA. [3]Emory National Primate Research Center, Emory University, Atlanta, GA 30329, USA. [4]Department of Psychiatry and Behavioral Sciences, Emory University, Atlanta, GA 30329, USA. [5]Kavli Institute for Systems Neuroscience, Norwegian University of Science and Technology, Trondheim, Norway. [6]These authors contributed equally: Brianna E. Hegarty, George W. Gruenhagen, Zachary V. Johnson. ✉e-mail: george.gruenhagen@gmail.com; todd.streelman@biology.gatech.edu

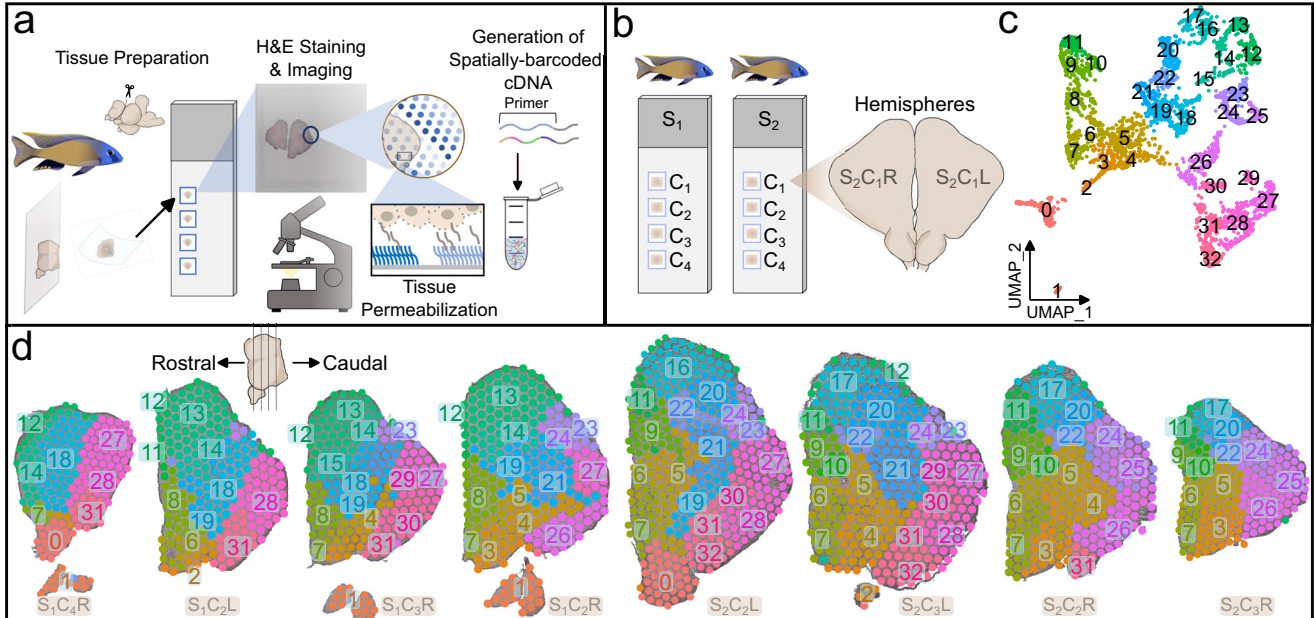

**Fig. 1 | Spatial transcriptomic profiling of the teleost telencephalon. a** Overview of spatial transcriptomics experimental pipeline. 10 μm coronal sections from cryo-preserved MC telencephala were placed on capture areas of 10x Genomics Visium slides and subsequently fixed, H&E stained, and imaged. Tissue was permeabilized for mRNA capture by barcoded primers on 50μm-diameter spots and spatially-barcoded cDNA was processed for downstream library preparation and sequencing. **b** Schematic of experimental design including four tissue sections on capture areas ($C_{1-4}$) from two male MC subjects (subject 1, $S_1$; subject 2, $S_2$), totaling eight samples processed on the 10x Visium platform. Tissue from the left and right telencephalic hemispheres (L and R) were visualized separately. **c** Unsupervised gene expression clustering of spots visualized in UMAP space. D. 50 μm-diameter spots on select tissue section hemispheres are colored by clusters from (**d**) The order of tissue sections along the rostrocaudal axis was determined through visual inspection of H&E-stained tissue.

## Results

### Spatially resolved gene expression profiles in the teleost brain

To investigate the anatomical landscape of the adult *Mchenga conophoros* (MC) telencephalon, we generated ST data using the 10x Genomics Visium platform (Fig. 1a). We dissected the telencephala from two adult male MCs (Supplementary Data 1), collecting four representative 10 μm coronal sections along the rostrocaudal axis per subject on the individual capture areas of 10x Visium slides (Fig. 1b; Fig. S1). Seven total sections ($n = 3$ from subject 1, $n = 4$ from subject 2) were chosen for downstream analysis based on tissue quality (Fig. S1). Due to rostrocaudal variation between the left and right telencephalic hemispheres on individual capture areas, we visualized and analyzed each hemisphere separately (Fig. 1b). In total, >500 million RNA reads (68 ± 2 million reads per capture area) were sequenced and aligned to the cichlid *Maylandia zebra* reference genome[16]. A total of 3971 spots (567 ± 65 spots per capture area) contained tissue and at least one unique molecular identifier (UMI), averaging 9729 ± 81 UMIs (range: 73–37,124) and 3639 ± 20 genes per spot (range: 73–7263; Fig. S2). Clustering parameters were chosen systematically using ChooseR, which evaluates clustering quality based on robustness metrics of bootstraps (Fig. S3; Supplementary Data 2). Using these near optimal parameters, 33 clusters of spots were identified (Fig. 1c, d). Marker genes of clusters were determined (Supplementary Data 3) and among these were canonical markers of well-defined anatomical regions in the teleost telencephalon (Fig. S4, Supplementary Note 1).

### Clusters correspond with anatomical regions

Anatomical regions were manually annotated based on tissue cytoarchitecture and expression of select canonical gene markers (Fig. 2a; Supplementary Note 1) and the distribution of clusters across these regions was assessed (Fig. S5). We observed high correspondence between anatomical regions and clusters (Rand index = 0.9487354), suggesting that transcriptional profiles of spots are strongly associated with their respective brain regions of origin.

The teleost telencephalon is subdivided into pallial (dorsal, D) and subpallial (ventral, V) domains which are predominantly populated by glutamatergic or GABAergic neurons, respectively. We identified these divisions in ST data based on expression patterns of conserved pallial (*bhlhe22, neurod1, neurod6b, eomesa*) and subpallial (*dlx2, dlx5*) marker genes (Fig. 2b; Fig. S6; Supplementary Note 1)[12,17,18]. As expected, we observed strong glutamatergic marker gene expression in pallial regions (*slc17a6, slc17a7a*) and GABAergic expression dominating subpallial regions (*gad1, gad2*; Fig. 2b; Fig. S6; Supplementary Note 1)[19].

Gene expression properties of profiled anatomical regions aligned with patterns described in previous teleost studies, briefly summarized below. Subpallial regions Vv (ventral nucleus of V) and Vl (lateral nucleus of V) are considered putatively homologous to the septal formation in mammals[1]. In these regions, we observed expression of markers in agreement with previous reports in fish (Vv: *isl1a, lhx6a, lhx8, and zic1*[17,20]; Vl: *sst1.1, npy* and *crhb*[1,21–23]; Fig. 2c, Fig. S6). The olfactory bulb granule cell layer (OB gc) expressed dopaminergic cell-type marker tyrosine hydroxylase (*th*)[24,25] along with *pax6*, a gene necessary for adult neurogenesis of dopaminergic neurons in the mammalian OB (Fig. 2c; Figs. S6–7)[26]. The dorsal, supra-commissural, and central nuclei of V (Vd, Vs, and Vc, respectively) expressed markers of mammalian striatal medium spiny neurons (MSNs; *penka, meis2, six3a*; Fig. 2c, Fig. S6)[27,28] and genes involved in dopaminergic transmission (*slc18a2, slc6a3*; Fig. S6)[24], in agreement with previously reported similarities between these regions and the mammalian striatum[1,29]. Teleost pallial regions Dl (lateral division of D) and Dm (medial division of D) are considered the putative homologues of the mammalian medial and ventral pallium, respectively[30]. In zebrafish, parvalbumin expression in the pallium has been used to establish histogenetic units, as it is observed in the Dl but not the Dm[31]. Interestingly, this pattern is not replicated in our data, as both Dl and Dm expressed *pvalb7* (Fig. 2c; Fig. S6), though this result may be due to differences in RNA versus protein expression. Urocortin (*ucn/uts1*) was highly expressed throughout the pallium, consistent with previous observations in teleosts (Fig. 2c; Fig. S6)[22,32].

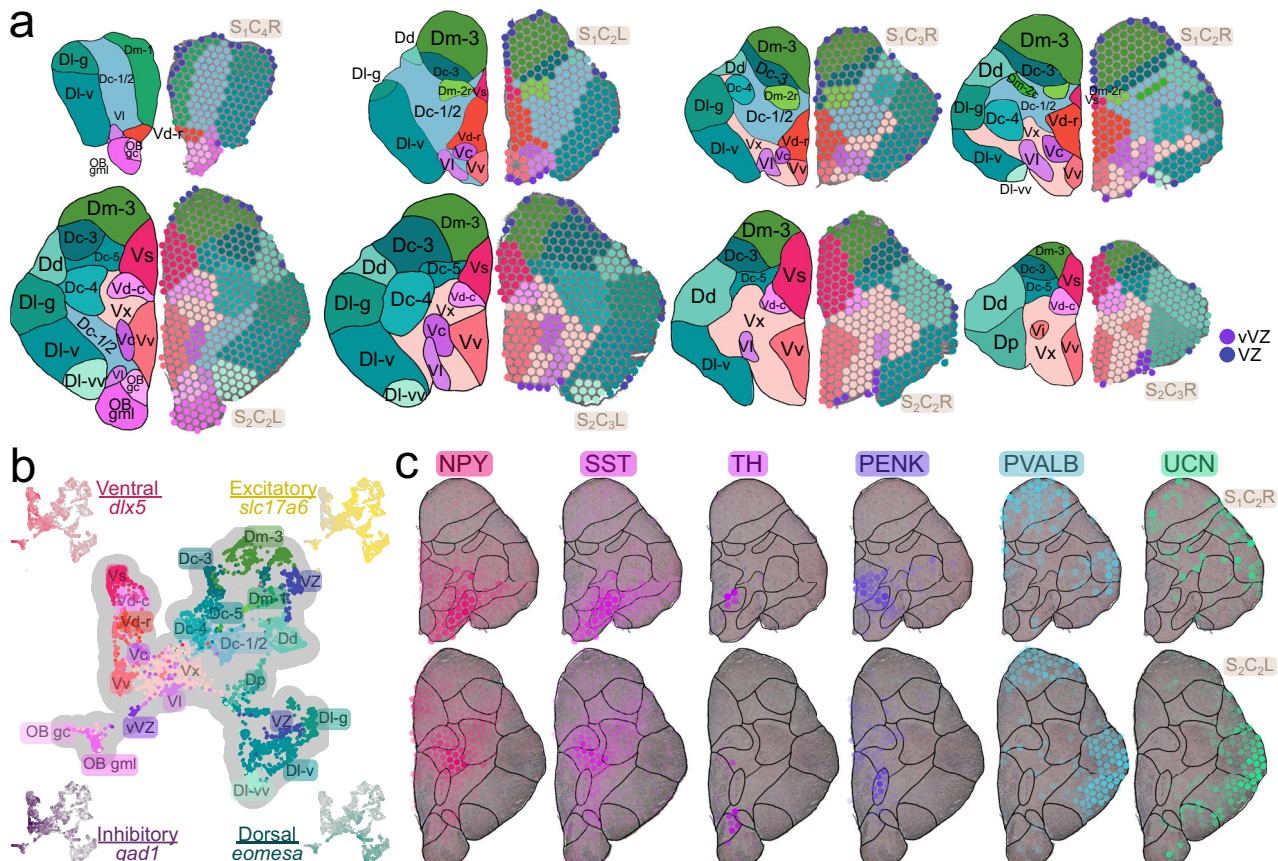

**Fig. 2 | Gene expression patterns in neuroanatomical regions. a** Anatomical annotation of select tissue sections (order same as Fig. 1) with representative coronal atlases shown on the left and spots colored by anatomical region on the right. Dc=central division of D, Dc-1,2,3,4,5=subdivisions of Dc, Dd=dorsal division of D, Dl=lateral division of D, Dl-d=dorsal subdivision of Dl, Dl-g=granular zone of Dl, Dl-v=ventral subdivision of Dl, Dl-vv=ventral zone of Dl-v, Dm=medial division of D, Dm-1,2,3=subdivisions of Dm, Dm-2c=caudal part of Dm-2, Dm-2r=rostral part of Dm-2, Dp=posterior division of D, OB gc=olfactory bulb granule (internal) cell layer, OB gml=olfactory bulb glomerular and mitral (external) cell layers,

Vc=central nucleus of V, Vd=dorsal nucleus of V, Vd-c=caudal part of Vd, Vd-r=rostral part of Vd, Vi=intermediate nucleus of V, Vl=lateral nucleus of V, Vs=supracommissural nucleus of V, Vv=ventral nucleus of V, Vx=unassigned subdivision of V, VZ=ventricular zone, vVZ=ventral ventricular zone, ON=olfactory nerve. **b** Spots visualized in UMAP space with anatomical annotations. Also shown is the expression of ventral (*dlx5*), dorsal (*bhlhe22*), excitatory (*slc17a6*), and inhibitory (*gad1*) marker genes. **c** Expression of neuromodulatory genes in representative section hemispheres, with transparency of spots scaled by expression level (spots not expressing the gene are completely transparent).

The majority of teleost gene expression studies have been limited by the necessary identification of gene targets prior to experimentation. The advent of ST technology eliminates this necessity, allowing anatomically-resolved profiling of thousands of genes simultaneously. In addition to replicating previously reported gene expression patterns, ST allowed us to identify novel marker genes for specific brain regions (Supplementary Data 4, Fig. S6).

### Anatomical distribution of cell-types in the telencephalon

The 10x Visium platform is a powerful tool for investigating the expression of thousands of genes within tissue anatomy, but lacks precise cellular resolution as each 50 µm-diameter spot may capture RNA from multiple cells and cell-types. Algorithms for cell-type deconvolution of spots address this limitation by estimating the cellular composition of individual spots. We recently profiled the cell-types in the MC telencephalon (38 subjects and 33,674 nuclei) using snRNA-seq[15]. Here, we predicted the anatomical location of these cell-types in our ST data using cell2location[33] (Fig. 3a, b), a top performing tool for cell-type deconvolution of spatial spots[34]. Spots were predicted to contain on average 8.7 ± 0.086 cells (Fig. S8; Supplementary Data 5, Supplementary Note 2) and most (>92%) were composed of more than one cell-type (4.3 ± 0.032 cell-types per spot on average). In general, anatomical locations of snRNA-seq cell-types (Fig. 3a, b) agreed with predictions based on previously-reported gene expression patterns in teleosts.

For example, 5_GABA cell-types, which demonstrated expression of well-known teleost OB marker genes (e.g., *th*, *pax6*)[24–26], mapped to the OB (Fig. 3c) and 4_GABA cell-types which expressed genes enriched in striatal MSNs (*sp9*, *six3a*)[35] were putatively located in the Vd and Vs regions (Fig. 3c). Our ST data not only represents a valuable resource for investigating the anatomical distributions of genes and cell-types in MC and other closely related cichlid species, but it can additionally serve as a teleost reference for deeper comparative neuroscience investigations.

### Conserved telencephalic cell-types in teleosts and mammals

Around 350–450 million years have passed since the last common ancestor of teleost fish and mammals[36]. Despite a vast array of studies aimed at disentangling the putative mammalian homologues of forebrain regions in fish, obstacles such as the 'everted' teleost pallium have made this challenging. To investigate the evolution of cell-types within the telencephalon, we analyzed conserved transcriptional signatures present in comparable cichlid[15] and mouse[37] forebrain datasets. To conduct these comparisons, previous studies[7–9] have correlated the expression of common one-to-one marker genes. Recently, a novel integrative approach, SAMap, has been designed for comparisons of cell-types from distantly related species, accounting for protein sequence divergence. We found that SAMap performed better when comparing cell-types from a downsampled dataset to the original underlying dataset[37] (SAMap $R^2$ = 0.436; Pearson's correlation;

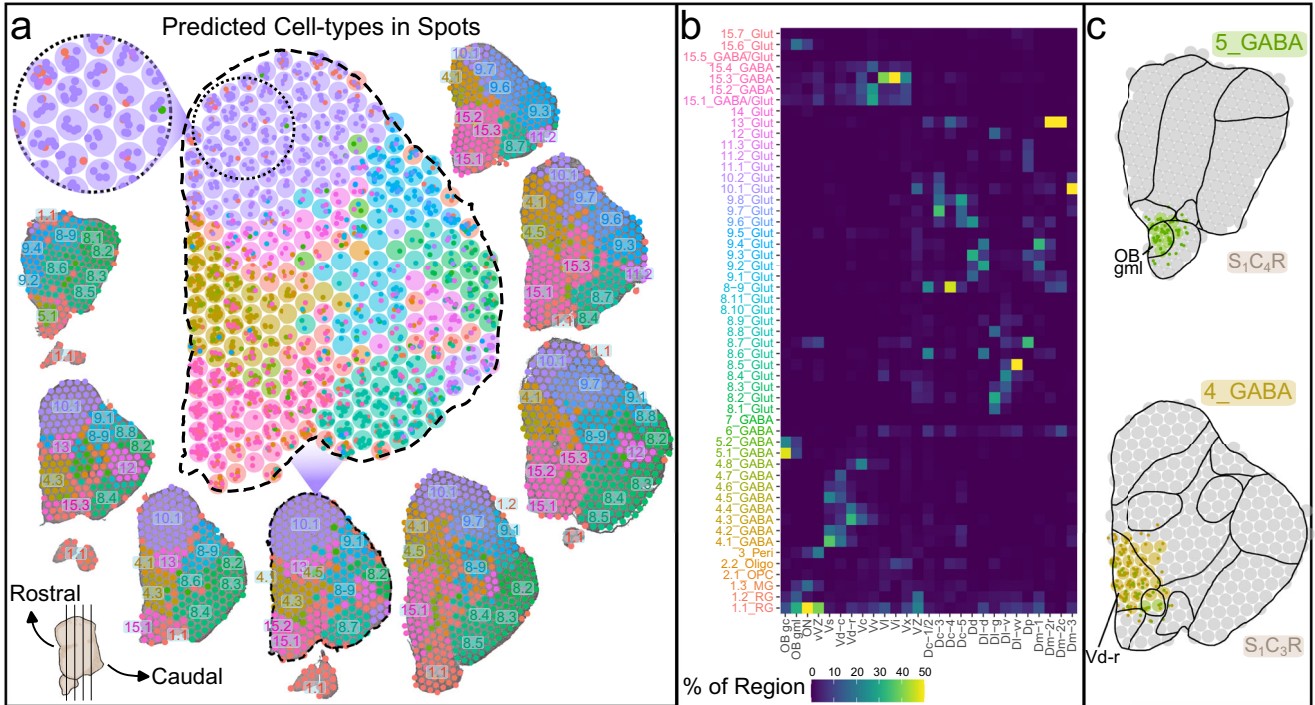

**Fig. 3 | Spatial context of cell-types profiled by snRNA-seq. a** Estimates of snRNA-seq cell-types[15] in ST spots by cell2location. Representative tissue hemispheres are ordered counter-clockwise as in Fig. 1 and each spot on the tissue is color-coded by the cell-type with the greatest number of predicted cells within the spot (color-coding scheme is shown in **b**). Cell-type abundance estimates in a representative tissue hemisphere ($S_1C_2R$) are shown with estimated cells as small dots within spots (larger circles). **b** Cell-type composition of annotated anatomical regions (color represents proportion of cell-types composing anatomical regions). Naming convention for snRNA-seq cell-types is outlined in Johnson et al. 2023[15]. **c** Anatomical distribution of 5_GABA and 4_GABA cell-types in $S_1C_4R$ and $S_1C_3R$ respectively. Spots are colored by the 5_GABA and 4_GABA cell-type with the greatest number of predicted cells within the spot and are otherwise colored gray. Small dots within spots represented predicted 5_GABA and 4_GABA cells.

Fig. 4a) and was therefore primarily used to compare cichlid and mouse telencephalic cell-types (Fig. S9, Supplementary Note 3). Sequence similarity-aware integration of these datasets projected nuclei/cells together in UMAP space (Fig. 4b), despite the vast evolutionary distances between these species. Next, a similarity score was calculated based on the mean k-nearest neighbors between cichlid and mouse cell-types and the score was compared to permutations (Fig. 4c; Supplementary Data 6). Cell-type pairs with similarity scores greater than all permutations ($q_{perm} = 0$; $n_{perm} = 1000$) are outlined below, with particular attention paid to pairs where both cell-types were reciprocal top hits.

We found strong cross-species correspondence of major cell-type classes, with glutamatergic, GABAergic, and non-neuronal populations demonstrating significant transcriptional similarities between cichlid and mouse (Fig. 4c). For example, strong correspondence was observed between oligodendrocyte cell-types (cichlid *plp1b* + 2.2_Oligo with mouse *Plp1* + MFOL1) and oligodendrocyte precursor cell-types (cichlid *olig2* + 2.1_OPC with mouse *Olig2* + COP1). Cichlid *fabp7* + /*slc1a3a* + radial glia populations (1.1_RG and 1.2_RG) resembled a mouse astrocyte cell-type (*Fabp7* + /*Slc1a3* + ACTE1) and 1.1_RG additionally bore similarity to mouse dentate gyrus radial glia (RGDG). In the teleost telencephalon, radial glia line the pallial and subpallial ventricular zones (VZ), which resemble the main neurogenic niches of the adult mammalian brain, the dentate gyrus, and subventricular zone[38]. In teleosts, radial glia function as stem cells, giving rise to new neurons and glia throughout adulthood[38] and are considered astroglial due to their expression of classical astrocyte markers[39] despite lacking the stellate morphology characteristic of mammalian astrocytes. Additionally, the teleost pallial VZ contains a population of non-glial proliferative progenitors largely composed of neuroblasts[40]. Cichlid 9.5_Glut, putatively concentrated along the pallial VZ, expressed markers of neuroblasts (*sox4*, *sox11*, *mex3a*, and *zeb2*) and bore molecular similarity to *Sox4* + granule neuroblasts of the hippocampal dentate gyrus (DGNBL1).

The most abundant neuroblast population in the teleost telencephalon forms a rostral migratory stream-like structure in the subpallial VZ, migrating into the OB[38] similar to observations in the subventricular zone of rodents[41]. Indeed, the cichlid cell-type which mapped to the glomerular/mitral layer (gml) of the OB (*eomesa* + /*tbx21* + 15.6_Glut) resembled mouse glutamatergic OB neuroblast populations (*Eomes* + OBNBL1). Cichlid *tp73* + 15.7_Glut and *cacna2d2* + 14_Glut, which were similar to several mouse neuroblast populations, additionally resembled dentate gyrus *Trp73* + /*Cacna2d2* + Cajal-Retzius cells. In the mouse brain, these cells play critical roles in hippocampal and neocortical development and radial glia migration[42].

In the mammalian brain, GABAergic neuronal lineages are derived largely from ganglionic eminence progenitor domains (e.g., medial ganglionic eminence, MGE; lateral ganglionic eminence, LGE). Early subpallial territories in the embryonic teleost brain have been proposed as MGE/pallidal-like or LGE/striatal-like based on regionalized expression of neuroregulatory genes[42]. We observed a reciprocal top hit between cichlid *six3a* + 4.2_GABA and a mouse LGE-derived striatal MSN population, *Six3* + MSN2 (Fig. 4c). In addition, several other cichlid 4_GABA cell-types, which mapped to the Vs and Vd, were significantly similar to mouse LGE-derived populations (MSN1-4). Consistent with our results, recent cross-species comparisons between the teleost and mouse forebrain also revealed strong correspondence between mammalian MSNs and subpallial cell-types expected to be located near the Vd in both zebrafish[12] and goldfish[23]. We found that LGE-derived cell-types putatively located in the cichlid and mouse OB mapped to each other with great specificity (*etv1* + 5.2_GABA to *Etv1* + OBDOP2). Additionally, we observed strong molecular conservation between cichlid GABAergic cell-types and mouse MGE-derived inhibitory interneurons, including *sst1.1* +, *npy* + 15.3_GABA, putatively Vl-derived, with mouse Sst-class interneurons (TEINH21) and between *pvalb6* + /*nxph1* + 6_GABA with mouse interneurons in the

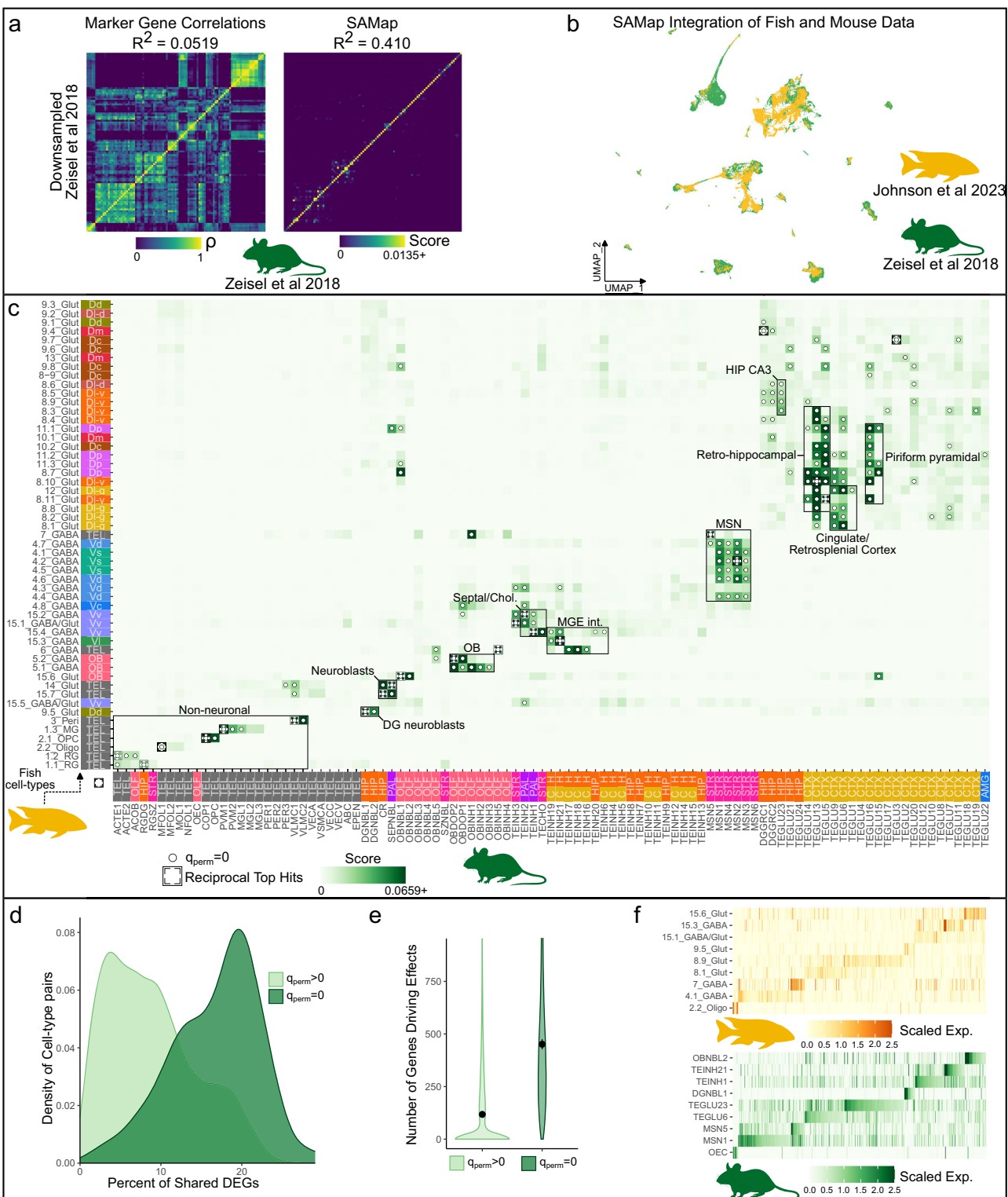

**Fig. 4 | Comparative analysis of cell-types in teleosts and mice. a** A direct comparison of methods for comparing distantly related scRNA-seq datasets suggested SAMap had greater accuracy than correlating shared marker genes (see Methods). **b** SAMap projects telencephalic cichlid nuclei ($n = 33,674$)[15] and mouse cells ($n = 100,531$)[37] together in UMAP space (cichlid nuclei are ordered to the front). **c** Similarity scores between cichlid (y-axis) and mouse cell-types (x-axis) with crosses denote reciprocal top hits and dots denoting significance determined by permutation testing ($q_{perm} = 0$, $n_{perm} = 1000$). Colored axis labels represent the predicted/probable region of origin of nuclei/cells. AMG=amygdala, CTX=cortex, OLF=olfactory, HIP=hippocampus, PAL=pallidum, STR=striatum, TEL=telencephalon. **d** Distribution of the percent of conserved markers for significant ($q_{perm}=0$, dark green) and non-significant ($q_{perm} > 0$, light green) cell-type pairs. The former had significantly more conserved marker genes than the latter (Welch Two Sample $t$ test; $p = 3.79e\text{-}49$, $t = 19.6$). **e** Violin plot of the number of genes driving effects for significant (dark green) versus non-significant (light green) cell-type pairs. Dots indicate the mean value and vertical lines represent the standard error. Possible outliers (<1st or >99th percentile) are not visualized. Cell-type pairs with significant similarities also had significantly more genes driving effects than other cell-type pairs (Welch Two Sample $t$ test; $p = 2.13e\text{-}30$, $t = 13.8$). **f** Scaled expression of DEGs driving select cell-type relationships ($n_{genes} = 529$). Cell-type pairs with significant similarity are in the same order on the y-axis and genes driving these relationships are in the same order on the x-axis.

hippocampus and cortex ($Pvalb+$, $Nxph1+$ TEINH17-18). A similar finding was reported by ref. 23, which found that goldfish cell-types located in the lateroventral subpallium (referred to by authors as the $Vsst$) demonstrated strong transcriptional similarity to mouse SST interneurons. Finally, several cichlid cell-types which mapped to the Vv resembled mouse cell-types of the septal nuclei ($lhx6a+$ 15.4_GABA with $Lhx6+$ TEINH1; $zic1+$ 15.2_GABA with $Zic1+$ TEINH2).

Due to the abundance of glutamatergic cell-types and their relative similarity, below we describe statistically significant cell-type relationships across species with less attention to reciprocity. Multiple populations which mapped to the Dm (10_Glut) and Dp (posterior division of D; 11_Glut and 8.7_Glut) bore similarity to a mouse cell-type in cortical entorhinal superficial layers (TEGLU5) as well as to piriform pyramidal cells (TEGLU15-17). Cichlid $bhlhe22+$ 8_Glut cell-types putatively located in the ventral zone of the lateral subdivision of D (Dl-v), referred to here as Dl-vv (8.5_Glut and 8.9_Glut) were transcriptionally similar to a mouse hippocampal CA3 cell-type ($Bhlhe22+$ TEGLU23; Fig. 4c). We also observed similarity between several $cckb+$ 8_Glut cell-types which mapped to the Dl-v (8.3_Glut, 8.4_Glut, 8.10_Glut, 8.11_Glut) and mouse cell-types of the subiculum (TEGLU13-14), a mammalian structure in the retrohippocampal formation that mediates hippocampal-cortical communication[43]. In line with this result, recent studies in zebrafish[12] and goldfish[23] also reported molecular similarity between the mouse subiculum and cell-types located in the Dl-v. Notably, predicted Dl-g (granular zone of Dl) cichlid $rbfox3b+$ cell-types, including 8.1_Glut and 12_Glut (Fig. S10), were transcriptionally similar to excitatory cortical projection neurons from cingulate/retrosplenial areas ($Rbfox3+$ TEGLU6 and TEGLU9). The robustness of these results was supported by analyses of shared marker genes and SAMap driving genes (Fig. 4d–f, Supplementary Note 4, Supplementary Data 7) and by additional comparisons performed using other teleost (goldfish) and mouse telencephalic datasets (Supplementary Notes 5, 6, Figs. S11, 12). Remarkably, the predicted dorsal-to-ventral organization of teleost neuronal populations that were transcriptionally similar to mammalian retrosplenial/cingulate, subiculum, and CA3 neuronal populations reflected the arrangement of these brain regions in mammals[44]. Taken together, our analyses reveal a suite of glutamatergic neuronal populations in teleosts bearing transcriptional and anatomical similarities to those populating mammalian cortical regions.

## Conserved telencephalic cell-types in teleosts and tetrapods

Across vertebrate lineages, the telencephalon demonstrates both commonalities shared between taxa as well as marked, specialized differences and many hypothesized brain region homologies in non-mammalian vertebrates are unclear and actively debated. To further investigate conserved telencephalic populations in vertebrates, we performed cross-species comparisons between cichlid cell-types and cell-types from tetrapod forebrain regions (Fig. 5a–c; Supplementary Data 8), including the axolotl telencephalon[9], turtle pallium[7], and songbird HVC, RA, and Area X regions[8]. The HVC and RA are involved in songbird vocal circuits and located in the DVR, a sauropsid pallial structure[8]. Following the same approach previously described, SAMap was used to integrate cichlid nuclei with datasets from each species separately, revealing several consistent patterns outlined below.

Our comparisons supported correspondence of major non-neuronal cell-types, including cichlid astroglia-like radial glia cell-types ($fabp7+$ / $slc1a2+$ 1.1_RGC and 1.2_RGC) to axolotl ependymoglia (Fig. 5a) and songbird astrocytes (Fig. 5c). Cichlid microglia (1.3_MG; $csf1r+$) and oligodendrocytes (2.2_Oligo; $plp1a+$) consistently mapped to their counterparts in all vertebrate comparisons (Fig. 5a–c) and oligodendrocyte precursor cells (2.1_OPC, $olig2+$) showed one-to-one mapping in songbirds. As with the comparison to the mouse telencephalon, $sox4+$ 9.5_Glut demonstrated significant similarity to neuroblasts in all comparisons. Taken together, these findings point towards strongly conserved molecular signatures present in major non-neuronal populations across vertebrates.

Cichlid GABAergic cell-types demonstrated consistent patterns across comparisons with other vertebrates (Fig. 5a–c; Fig. S13). Cichlid 5_GABA cell-types that mapped to the OB resembled olfactory LGE-derived cell-types (axolotl GABA1 and GABA3; turtle i01) and songbird GABA-1-1, which bore similarity to mammalian non-neocortical LGE-class neurons according to previous analysis[8]. Cichlid $meis2+$, $six3a+$ 4_GABA cell-types similar to mouse striatal MSNs also resembled songbird MSNs from Area X and LGE-derived cell-types in axolotl (GABA11) and turtle (i05). In line with results from our cichlid-mouse analysis, we observed strong transcriptional similarity between vertebrate MGE-derived populations and cichlid $pvalb6+$, $nxph1+$ 6_GABA (turtle i12; songbird GABA-3; axolotl GABA2 and GABA6) and $sst1.1+$, $npy+$ 15.3_GABA (axolotl GABA17 non-reciprocally; turtle i10; songbird GABA-2).

Additionally, we found evidence of conserved molecular features in several glutamatergic cell-types (Fig. 5a–c). Similar to our cichlid-mouse comparison, a subset of 8_Glut cell-types predicted to populate the Dl-v (8.5_Glut and 8.9_Glut) consistently grouped with hippocampal-like cell-types in vertebrate datasets, including reciprocal top hits between cichlid 8.5_Glut and turtle e34 from the dorsal medial cortex (DMC) and cichlid 8.9_Glut with axolotl GLUT7. Cichlid Dc (central division of D) cell-type 9.8_Glut was a reciprocal top hit with a turtle population in the posterior dorsal ventricular ridge (pDVR; e25), reported as similar to the mammalian pallial amygdala[7]. Other cichlid cell-types in the Dc and neighboring Dm-2r (rostral part of subdivision 2 of Dm; 13_Glut) were transcriptionally similar to the pDVR. A cell-type putatively located in the Dp (11.3_Glut) strongly resembled a turtle cell-types in the anterior lateral cortex (aLC; e12), the putative reptilian homologue of the piriform cortex[7]. Two cichlid cell-types that mapped to the Dl-g had reciprocal top hits with populations in the turtle anterior dorsal cortex (aDC; 8.2_Glut with e16; 12_Glut with e14), and a third cichlid cell-type was also transcriptionally similar to an aDC cell-type (8.8_Glut with e16). One of these cichlid cell-types (12_Glut) was also transcriptionally similar to a cell-type in the turtle anterior dorsal ventricular ridge (aDVR, 12_Glut with e01). Interestingly, several Dl-g cell-types were transcriptionally similar to axolotl cell-types that were previously determined to be transcriptionally similar to cell-types in turtle aDC and aDVR and to mouse cell types in retrosplenial/cingulate cortex and other cortical regions (8.1_Glut with axolotl GLUT2; 8.2_Glut with axolotl GLUT2, GLUT20; 12_Glut with axolotl GLUT22)[8,9]. These axolotl clusters were present in microdissections of dorsal pallium (GLUT2, GLUT20), medial pallium (GLUT2, GLUT20), and lateral (including ventral) pallium (GLUT2, GLUT20, GLUT22)[9].

The robustness of these relationships was evidenced by significant cross-species cell-type pairs exhibiting more shared marker genes (Welch Two Sample $t$ test; axolotl $p = 2.17\text{e-}10$; turtle $p = 5.54\text{e-}25$; songbird $p = 1.63\text{e-}19$; Fig. S14; Supplementary Data 9) and SAMap driving genes (Welch Two Sample; axolotl: $p = 1.79\text{e-}10$, $t = 7.08$; turtle: $p = 1.69\text{e-}11$, $t = 7.27$; songbird: $p = 5.68\text{e-}20$, $t = 10.49$; Fig. S14; Supplementary Data 9) compared to non-significant cell-type pairs (genes that defined the most transcriptionally conserved cell populations across vertebrates are provided in Figs. S15, 16; Supplementary Data 10). Taken together, our analyses reveal strong molecular conservation of forebrain cell-types across vertebrate lineages.

## Transcriptional similarities of anatomical regions between teleosts and tetrapods

Our atlas of the cichlid telencephalon provides a powerful tool to explore transcriptional similarities of telencephalic regions across species, amidst competing hypotheses about teleost-to-mammal brain region homologies. Using SAMap, we integrated ST cichlid data with anatomically-annotated scRNA-seq data from turtle pallial microdissections[7] and separately with ST data from the mouse brain[45]. We then used these anatomically-rooted transcriptional data to more directly compare evolutionary relationships of brain regions across vertebrate species. Below we outline significant cross-species relationships between brain regions with particular attention to

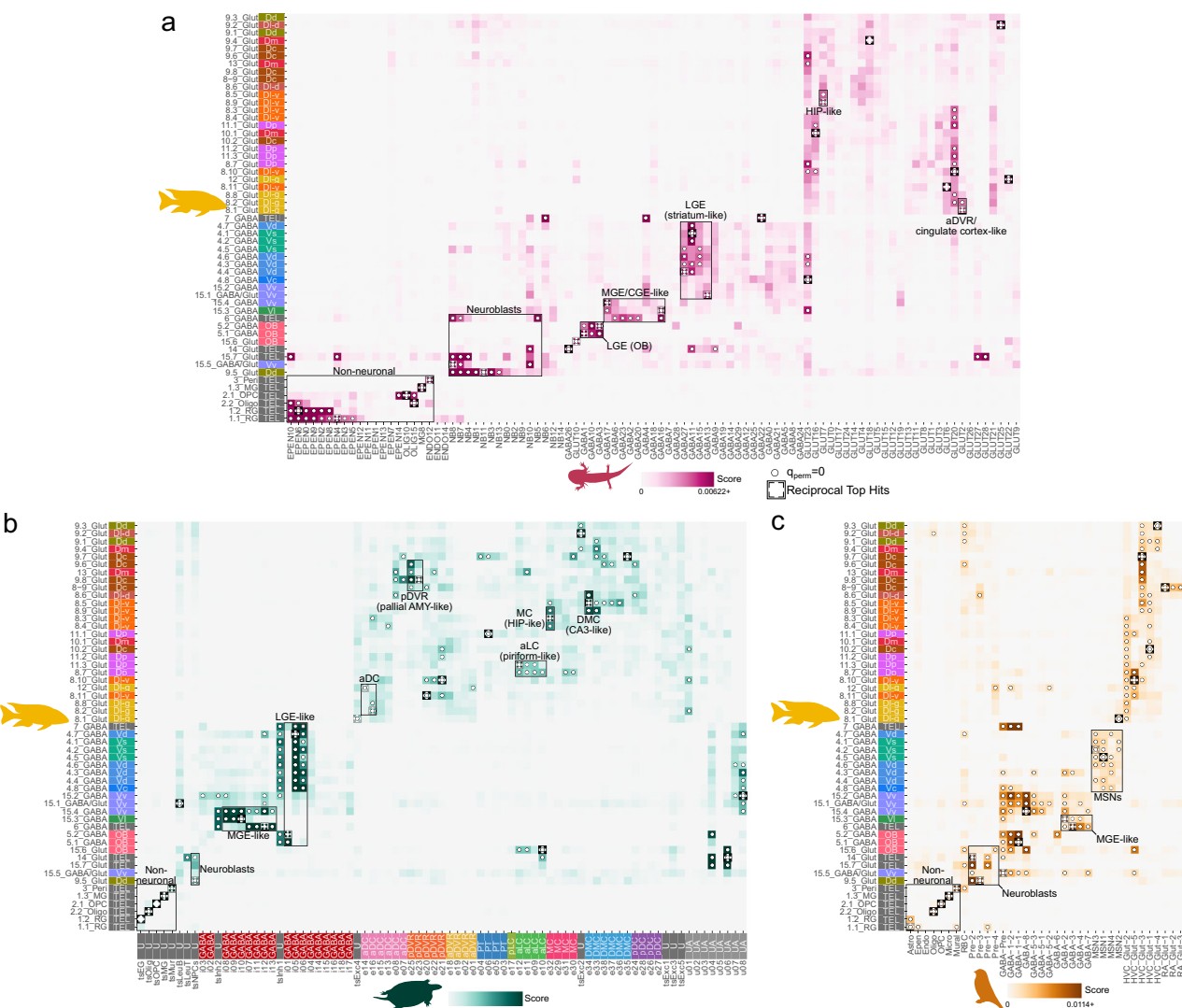

**Fig. 5 | Comparative analysis of cichlid telencephalic cell-types to axolotl, turtle, and songbird forebrain cell-types.** Pairwise transcriptional comparisons of cichlid telencephalic cell-types to axolotl[9] (**a**), turtle[7] (**b**), and songbird[8] (**c**) cell-types using SAMap. Cell-type pairs with higher similarity scores indicate greater transcriptional similarity, crosses denote reciprocal top hits, and dots denote similarity scores greater than all permutations ($q_{perm}$ = 0). Turtle cell-types are colored by their inferred region of origin with the following additional labels: U=unknown, UA=unassigned.

reciprocal top hits that were additionally supported by cell-type relationships described earlier.

Several cichlid brain regions were transcriptionally similar to turtle pallial brain regions: Dl-vv and the DMC, Dp and the aLC, Dm-2r and the pDVR, and the Dl-g and the aDVR (Fig. 6a; Supplementary Data 11). The relationship between the Dl-vv and the DMC was supported by our cell-type analysis: Dl-vv cell-type 8.5_Glut was a reciprocal top hit with a DMC cell-type (e34), and was additionally transcriptionally similar to several more (Fig. 5b) as well as to mouse hippocampal CA3 cell-type TEGLU23 (Fig. 4c), consistent with ref. 7. Similarly, the reptilian aLC and cichlid Dp have previously been reported as putative homologues of the mammalian olfactory (piriform) cortex[7,43] and their homology to each other was further supported by our cell-type comparison (cichlid 11.3_Glut with turtle e12; Fig. 5b). The relationship between Dm and pDVR was supported by cichlid cell-types putatively located in the ventromedial Dm (Dm-2r, Dm-2c (caudal part of subdivision 2 of Dm)) showing significant similarity to turtle pDVR cell-types (13_Glut with e25; Fig. 5b). Notably, both regions have been previously and independently compared to the mammalian pallial amygdala[7,46]. Interestingly, Dl-g was transcriptionally similar to turtle aDVR but not to turtle aDC. Although several cichlid Dl-g cell-types (8.2_Glut, 8.8_Glut, 12_Glut) were most transcriptionally similar to aDC cell-types

(e14 and e16), Dl-g cell-type 12_Glut was also transcriptionally similar to aDVR cell-type e01 (Fig. 5b).

These results largely aligned with comparisons between cichlid and mouse brain regions. Consistent with previous teleost studies, we observed strong similarities between cichlid and mouse subpallial regions: cichlid OB gc with the main olfactory bulb (MOB), the Vd with the striatum (Vd-r (rostral part of Vd) with nucleus accumbens, ACB), and the Vv with the lateral septal complex (LSX) (Fig. 6b; Supplementary Data 11). Interestingly, cichlid-mouse comparison of pallial regions revealed that the Dl-vv, a densely populated subregion at the ventral-most pole of Dl, bore strong similarity to the CA3 subregion of the hippocampus, consistent with previously proposed relationships between the teleost Dl-v and mammalian hippocampal pallium[5,43,46,47]. Expression of *prox1* in the zebrafish Dl-g has led to a proposed homology between this region and the mammalian hippocampal dentate gyrus[18], but this gene was not strongly expressed in cichlid Dl-g or other cell-types. Further, we did not find transcriptional similarity between Dl-g and mammalian dentate gyrus, but instead found that Dl-g was most transcriptionally similar to the mammalian visual cortex (VIS, Fig. 6b), complementing our earlier reported transcriptional similarities between Dl-g cell-types (8.1_Glut and 12_Glut) and were significantly similar to mouse neocortical cell-types (TEGLU6, TEGLU9; Fig. 4b).

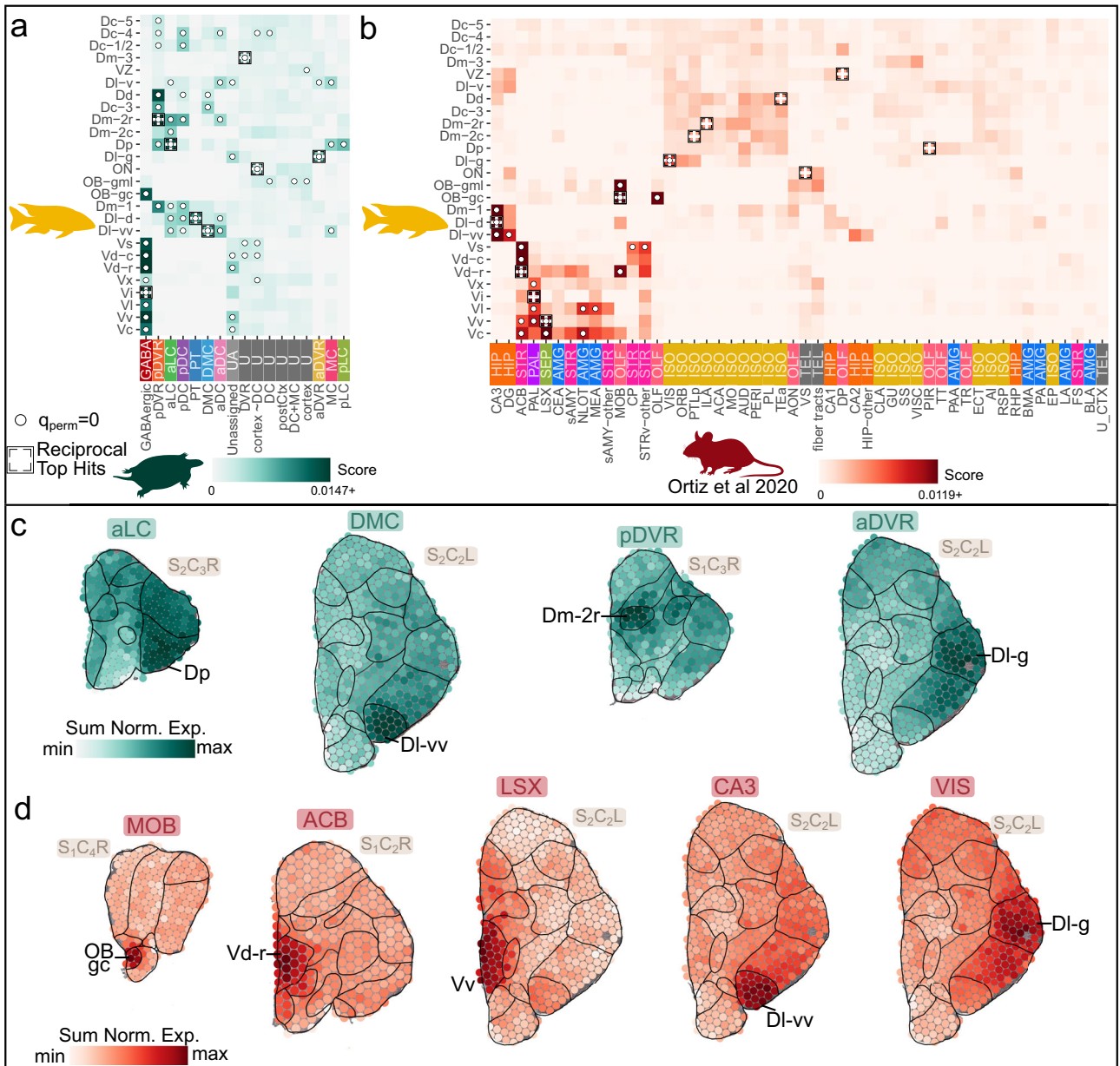

**Fig. 6 | Comparison of neuroanatomical regions in cichlids to turtles and mice.** **a** Transcriptional comparisons of annotated cichlid telencephalic ST anatomical regions to anatomical annotations of turtle pallial scRNA-seq data[7] using SAMap. Greater similarity scores indicate greater transcriptional similarity, crosses denote reciprocal top hits, and dots denote significance ($q_{perm} = 0$). **b** Transcriptional comparisons of annotated cichlid telencephalic ST anatomical regions to Allen Brain Atlas (ABA) region annotations of mouse ST data[45] using SAMap. AMG=Amygdala, HIP=Hippocampal formation, ISO=Isocortex, OLF=Olfactory areas,

STR=Striatum, PAL=Pallidum, SEP=Septum and TEL=Telencephalon. **c** Summed normalized expression of genes driving select significant relationships between cichlid and turtle anatomical regions (turtle aLC and cichlid Dp; turtle DMC and cichlid Dl-vv; turtle pDVR and cichlid Dm-2r; turtle aDVR and cichlid Dl-g). **d** Summed normalized expression of genes driving select significant relationships between cichlid and mouse anatomical regions (mouse MOB and cichlid OB gc; mouse ACB and cichlid Vd-r; mouse LSX and cichlid Vv; mouse CA3 and cichlid Dl-vv; mouse VIS and cichlid Dl-g).

Additionally, the cichlid Dp was most transcriptionally similar to the mouse piriform cortex (PIR), though it did not meet our significance threshold. This similarity agrees with the above comparisons to turtle brain regions and with the partial eversion model of the teleost telencephalon[2]. These relationships were reinforced by additional analyses of shared marker genes and SAMap driving genes (Fig. 6c, d, Fig. S17, Supplementary Notes 7, 8, Supplementary Data 12).

## Anatomical subregions in vertebrates exhibiting transcriptional similarity to the mammalian cortical regions

The teleost brain lacks a layered cortex and telencephalic populations that resemble those present in this elaborated mammalian structure remain a

matter of debate. A popular theory casts doubt on the existence of a cortex-like homologue in fish[5], while others propose the Dc[2,31] or the Dl-d(dorsal subdivision of D)/g region as putative phenotypic homologues of the dorsal pallium, which gives rise to the neocortex and mesocortex in mammals[2,3,48,49]. Similarly, the homologies of specific reptilian brain regions to mammalian cortical regions are the subjects of ongoing debate[7]. We find transcriptional similarity between predicted Dl-g cell-types to mouse neocortical populations (Fig. 4c) and cell-types from the turtle aDC and aDVR (Fig. 5b). Previous analyses found that the turtle aDC and aDVR possessed cell-types with significant correlations to the mammalian neocortex using all genes, but only cell-types in the aDC had significant correlations when the analysis was restricted to transcription factors[7]. To gain further insight into

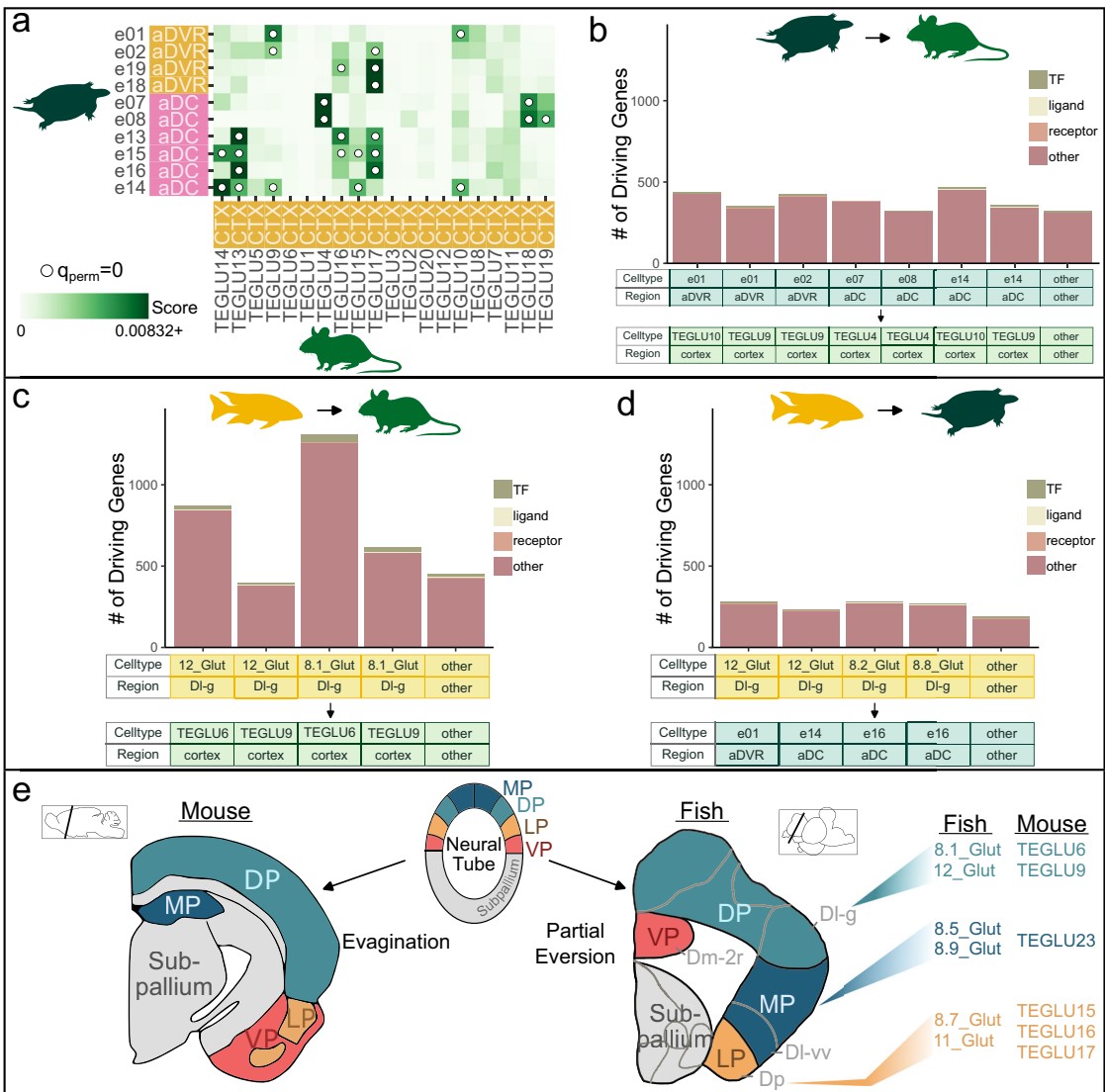

**Fig. 7 | Cortical signatures in reptiles and fish. a** Transcriptional comparison of cell-types in the turtle and mouse forebrain using SAMap. Results for cell-types from the turtle aDC and the aDVR[7] to cortical mouse cell-types[37] are shown (full results can be found in Fig. S18). **b–d** Composition of gene categories driving significant cichlid-mouse, cichlid-turtle, and turtle-mouse cell-type pairs (transcription factors, neuromodulatory ligands and receptors). No significant differences in composition were found.
**e** Simplified schematic demonstrating the arrangement of pallial zones, supported by our data and results, in representative coronal sections of the adult mouse (modified from

ref. [79]) and cichlid forebrain resulting from the developmental processes of neural tube evagination and eversion, respectively. Selected cichlid and mouse[37] cell-type pairs shown on the right demonstrate significant transcriptional similarity and support a model of partial eversion[2] of the teleost pallium. Colors represent the dorsal, medial, lateral, and ventral pallium divisions (separated by black lines). Teal=dorsal pallium (DP), red=ventral pallium (VP), navy blue=medial pallium (MP), orange=lateral pallium (LP) and gray=subpallium. Gray lines represent subregions of the teleost telencephalon.

the transcriptional relationships between these structures, we used SAMap to compare turtle[7] and mouse[37] scRNA-seq datasets.

This analysis revealed significant and reciprocal relationships between both an aDC cell-type (e07) and an aDVR cell-type (e01) to mouse neocortical cell populations (TEGLU4 and TEGLU9 respectively) (Fig. 7a; Fig. S18). While significant and reciprocal, these turtle cell-types had similarities to other mammalian cell-types that were not neocortical. The same is true of other significant, but non-reciprocal similarities were observed aDC and aDVR cell types and other mouse cortical and non-cortical populations (Fig. 7a; Fig. S18), consistent with previous reports that turtle aDC and aDVR contain cell-types that are transcriptionally similar to mammalian neocortex[7]. Interestingly, this previous work also found that the relationship between aDVR and neocortex, but not between aDC and neocortex, was no longer present when only transcription factors were analyzed, raising the possibility that some transcriptional similarities may be driven by specific gene classes (e.g., transcription factors). To investigate

this, we tested whether the composition of different gene classes (using curated lists[7,15] of transcription factors, neuromodulatory ligands, neuro-modulatory receptors, or other; see Methods; Supplementary Data 13) among SAMap driving genes differed across the different transcriptional relationships we identified among cichlid, turtle, and mouse brain regions. This analysis revealed that the composition of gene classes among SAMap driving genes did not differ across cell-type/brain region similarities for turtle-mouse comparisons, cichlid-mouse comparisons, or cichlid-turtle comparisons (Fig. 7b–d, Supplementary Data 13, Supplementary Note 9). Taken together, our results are largely consistent with previous work, but support the idea that cross-species cell-type and brain region transcriptional similarities are driven by diverse functional categories of genes.

Based on these results, together with our earlier findings and previous literature, we propose the following brain regions contain molecularly conserved cell-types in cichlids, turtles, and mice: Dl-g, aDC, and meso-cortex (retrosplenial/cingulate cortex, subiculum)[49]; Dl-vv, DMC, and

hippocampus (CA3); Dm-2r, pDVR, pallial amygdala; Dp, aLC, and piri-form cortex. These findings support the arrangement of putative pallial divisions outlined in the partial eversion model, which proposes the Dl-v as medial pallial, the ventromedial Dm as ventral pallial, the Dp as lateral pallial, and the dorsal Dl as dorsal pallial[2] (Fig. 7e).

## Discussion

Here we create a spatially-resolved molecular atlas of the *Mchenga conophoros* telencephalon using complementary techniques, snRNA-seq and ST, to investigate cell-types sequenced at single cell resolution within tissue architecture. We find remarkable correspondence between transcriptional profiling and neuroanatomy, whereby both ST clusters and snRNA-seq cell-types align strongly with well-described neuroanatomical subregions of the fish telencephalon, consistent with previous observations in the mouse brain[33,45]. Using a comparative approach that considers protein sequence similarity between species, we identify a suite of non-neuronal and neuronal populations that show conserved patterns of gene expression across all major vertebrate lineages. By comparing both cell-types and regions across species, we find support for widely-accepted as well as controversial evolutionary relationships between specific teleost and mammalian brain regions, including a putative teleost homolog for mammalian mesocortex.

Our results are consistent with previous work in goldfish[23] and zebrafish[12]. Although differences in dissections, clustering, and cell-type/region annotations between studies can introduce noise to cross-species mapping, our comparative analyses reveal strong correspondence of spatially-resolved cell-type atlases between cichlids and goldfish, including strong transcriptional conservation of GABAergic cell types mapping to similar subpallial domains in both species, aligning with a previous study of the zebrafish telencephalon[12]. Similarly, despite differences in anatomical annotations of the cichlid and goldfish pallium, a suite of glutamatergic cell-types also demonstrate strong transcriptional conservation. In cichlid, goldfish, and zebrafish studies, similarities were found between cell-types in the Dl-v to mammalian populations in the retrohippocampal formation, including the subiculum, a relationship we discuss more below. Future studies across the diversity of teleost lineages, incorporating ST, are needed to trace the evolution of cell-types and brain regions in this diverse group.

We find support for homologies between specific cichlid GABAergic populations and mammalian MGE-derived *Sst+* and *Pvalb+* interneuron classes and LGE-derived populations, including OB cell-types and striatal MSNs. Cichlid MSN-like cell-types, like transcriptionally similar goldfish MSN-like cell-types[23], were transcriptionally similar to mouse MSN cell types and anatomically mapped to the Vd, a region which receives substantial dopaminergic input and is considered homologous to the mammalian nucleus accumbens of the striatum[1,29]. Additionally, these cell-types were transcriptionally similar to goldfish GABAergic MSN-like cell-types which mapped to the Vd, further strengthening the comparison between this region in teleosts to the mammalian striatum. Our results also support a previously proposed homology between the cholinergic Vv and the tetrapod septum[50]. Various studies have shown that both the striatum and septum, and their putative homologues in fish, play important roles in mediating evolutionarily-relevant social behaviors[50–53]. Our results strongly suggest that core cell-types in these subpallial regions are conserved across vertebrates, perhaps due to essential behavioral functions.

Our results also support the partial eversion model for teleost pallial organization[2]. The pallium has presented perhaps the most challenging puzzle in vertebrate brain evolution due to its markedly unique adaptations across vertebrate clades. The teleost telencephalon exhibits an everted morphology, but the extent of eversion is unclear. Some models suggest a complete eversion of all pallial zones, while others propose only a partial eversion[2–6]. Consequently, these models differ in the teleostean equivalents of mammalian dorsal, lateral, and ventral pallium divisions, but agree that the medial (hippocampal) pallium is displaced to a dorsolateral location in fish[43]. Disagreement between eversion models is rooted partly in the caudolateral position of the Dp, which is the main pallial recipient of secondary olfactory input in the fish telencephalon[43] and is thus the putative

homologue of the mammalian lateral (olfactory) pallium. Indeed, studies in zebrafish reveal functional roles for the Dp in odor-evoked activity similar to those performed of the mammalian piriform cortex[54,55]. The altered position of the Dp relative to the Dl-v is accounted for in the partial eversion model[2]. In support of this model, we reveal transcriptional similarity between the cichlid Dp and its associated cell-types and the mammalian piriform cortex populations. Furthermore, these cichlid populations also resemble the turtle aLC and its corresponding cell-types, considered the reptilian homologue of the piriform cortex[7]. These data are compelling evidence for molecular conservation of pallial olfactory populations across vertebrates.

Support for the partial eversion model also arose from our analysis of Dm. In mammals, the ventral and dorsal pallial divisions give rise to the pallial amygdala and neocortex, respectively, both of which are highly complex structures whose homologues in other vertebrates have been subjects of intense debate. For example, the sauropsid dorsal ventricular ridge (DVR) has been proposed as a possible homologue of mammalian isocortex, but is a ventral pallial structure and therefore has been increasingly viewed as pallial amygdalar[56,57]. Similarly, the teleost Dm has been compared to both the dorsal and ventral pallial divisions of the tetrapod brain[2,58]. The latter stems in part from a demonstrated role of the Dm in emotional learning and behavior[58] and critically, these experiments largely targeted the ventromedial portion of Dm positioned along the midline[59]. Furthermore, connections of the ventromedial Dm are comparable to those of the pallial amygdala, whereas the dorsal Dm receives ascending sensory input from the preglomerular complex resembling mammalian thalamic-neocortical pathways[59]. The aforementioned partial-eversion model accounts for these observations, proposing the ventromedial Dm as ventral pallial and the remaining Dm as dorsal pallial. Consistent with this model, we find both cell-type and region transcriptional similarity between ventromedial Dm (Dm-2r) and turtle pDVR, which previous work found to be transcriptionally similar to the pallial amygdala[7].

The hippocampus, functions in memory, spatial navigation and learning[60] and is generally considered to be highly conserved across vertebrates, although specific subregion homologies are debated[61]. Behavioral studies in goldfish have supported the ventral subdivision of the dorsolateral pallium (Dl-v) as a possible hippocampal homolog, demonstrating that it is necessary for similar learning, memory, and behavioral functions as the mammalian hippocampus[47,62,63]. Our results support this homology, as the ventral-most Dl-v region and its corresponding cell-types exhibited striking transcriptional similarities to cell-types in mammalian hippocampal CA3 region, and we additionally observed transcriptional similarities between this region and both hippocampal CA3 and the dentate gyrus, both of which have been previously been implicated in social and spatial information processing[64,65].

Our results additionally support teleost homologs of mammalian mesocortical neuronal populations. In the mammalian brain, the retrosplenial cortex is positioned between the 6-layer neocortex and the hippocampal formation and forms reciprocal connections with the subiculum, the hippocampus, and cortical regions[66–68], including the visual cortex[69]. Due to its unique anatomical position, the subiculum and retrosplenial cortex are often classified as mesocortex[49], a transitional cortex that develops from the embryonic dorsomedial pallium[70]. We find strong transcriptional similarities between cichlid Dl-g/Dl-v cell-types and mammalian retrosplenial/subiculuar cell-types, and these relationships were further reinforced by comparisons to a second mouse brain scRNA-seq dataset[71]. Strikingly, we find that the anatomically sequential (dorsal-to-ventral) organization of populations transcriptionally similar to mouse retrosplenial cortical, subicular, and hippocampal populations reflects the arrangement of these regions in the mouse brain. Taken together, these results further support the partial eversion model, and support the idea that specific neuronal populations in teleost Dl-g/Dl-v and mammalian mesocortex are homologous.

The relationships between cichlid Dl and mammalian retrosplenial/subiculuar/hippocampal regions are particularly interesting given our recent work supporting a central role for Dl in cichlid bower-building behavior, a courtship behavior involving the spatial manipulation of sand[15].

Two cell-types spatially mapping to Dl-g and Dl-v (8.1_Glut and 8.4_Glut, respectively) exhibited building-associated changes in their relative proportions. Behavior-associated signatures in a radial glial subpopulation bordering Dl, together with cell-cell communication analyses, supported a circuit model in which multiple neuronal populations and glia together coordinate the cellular reorganization within Dl during building. Here we find transcriptional similarities between 8.1 Glut and a mouse retrosplenial/cingulate layer 2-derived cell population (TEGLU6), and between 8.4_Glut and a mouse subiculum-derived cell population (TEGLU13). Previous work demonstrates a functional role for the retrosplenial cortex and subiculum in mammalian spatial/sociospatial learning and memory[72,73]. In light of these observations and our results, it is intriguing to speculate that bower-building is regulated by conserved mesocortical-like cell populations and circuits that encode representations of the social and spatial environment.

The evolutionary origins of the mammalian cortex have been intensely debated. One center of attention has been the evolutionary relationships between the aDC and DVR in reptiles and the mammalian cortex[7]. Interest in the DVR as a possible cortical homolog grew through repeated observations that it shared similar molecular, connective, and functional properties with the mammalian cortex[56,74]. Importantly, however, the mammalian cortex develops from the dorsal pallium, whereas the DVR develops from a distinct ventral pallial zone that exhibits divergent neurodevelopmental transcriptomic trajectories[10,75,76]. In contrast, the reptilian aDC arises from the dorsal pallium, and also bears striking molecular, connective, and functional similarities with the mammalian cortex[7]. For these reasons, the aDC is increasingly considered to share evolutionary origins with the mammalian cortex[57]. Interestingly, our analyses echoed the cortical signatures of both regions. Although our cross-species cell-type and region mapping usually converged, one exception was that Dl-g cell-types were more transcriptionally similar to aDC (turtle) and retrosplenial (mouse) cell-types, whereas the Dl-g region was more transcriptionally similar to aDVR (turtle) and visual cortex (mouse). Factors that may cause divergence between cell-type and region mapping include the evolution of distinct cell-types within a brain region in one or both lineages, variation in how brain regions and subregions are defined in relation to lineage-specific patterns of cellular organization, and technical differences across studies (e.g., sequencing methodology, region annotation, etc.). In addition to cell-type mapping, the immediate proximity between Dl-g and other hippocampal-like regions and cell-types mirrors the immediate proximity between aDC and other hippocampal-like regions and cell-types in reptiles, and furthermore is consistent with the partial eversion model. Taken together, these results support cichlid Dl-g as dorsal pallial, although striking transcriptional similarities to the reptilian ventral pallial aDVR are apparent, consistent with previous work[7,8,10].

In summary, our work demonstrates deep cell-type and region homologies in the vertebrate forebrain and addresses several knowledge gaps surrounding vertebrate brain evolution. Perhaps most notably, we find support for a partially everted telencephalon in teleosts compared to mammals, a pallial amygdala-like ventral Dm subdivision, and a dorsal pallial Dl-g region containing mesocortical-like neuronal populations similar to those populating the reptilian aDC and the mammalian subiculum and retrosplenial/cingulate cortex. Future comparative studies will deepen our understanding of cell-type and brain region evolution among teleosts and across vertebrate lineages.

## Methods
### Subjects
Male *Mchenga conophoros* cichlids used in this study were fertilized and raised into adulthood (>180 days) in the Georgia Institute of Technology (Atlanta, GA) Engineered Biosystems Building cichlid aquaculture facilities. All procedures were approved by the Institute Animal Care and Use Committee at the Georgia Institute of Technology (IACUC protocol number A100029). We have complied with all relevant ethical regulations for animal use. All animals were collected as fry from mouthbrooding females approximately 14 days post-fertilization and raised on a ZebTec

Active Blue Stand Alone system until approximately 60 days post-fertilization, at which point animals were transferred to 190-L (92 cm long × 46 cm wide × 42 cm tall) glass aquaria 'home tanks' maintained on a central recirculating system. Environmental conditions of aquaria were as follows: subjects were housed in pH = 8.2, 26.7 °C water in social communities (20–30 mixed-sex individuals) and maintained on a 12-h:12-h light:dark cycle (full lights on between 8 am and 6 pm Eastern Standard Time (EST)) and dim lights for 60 min periods between light-dark transition (7 am–8 am and 6 pm–7 pm EST). Subjects were fed twice daily between 8–9 am and 2–3 pm (Spirulina Flake; Pentair Aquatic Ecosystems, Apopka, FL, U.S.A.). Reproductive adult subject males were introduced from home tanks to experimental tanks as described in ref. 15 which contained sand and four reproductive adult size-matched stimulus females of the same species.

### Tissue processing
Adult subject males ($n = 2$) were collected from experimental tanks between 11 am and 2 pm EST (3–5 h after full lights-on) to control for potential effects of food intake and circadian timing on brain gene expression. Subjects were rapidly anesthetized with tricaine immediately following collection, measured for body mass (BM) and standard length (SL), and decapitated for brain extraction. Telencephala were dissected under a dissection microscope (Zeiss Stemi DV4 Stereo Microscope 8x - 32x, 000000-1018-455), in Hibernate AB Complete nutrient medium (HAB; with 2% B27 and 0.5 mM Glutamax; BrainBits) containing 0.2 U/μl RNase Inhibitor (Sigma) to prevent RNA degradation. Immediately following dissection, telencephala were embedded in disposable base cryomolds (7 mm × 7 mm × 5 mm; Simport Scientific) containing chilled Optimal Cutting Temperature (OCT) compound (TissueTek Sakura) and flash-frozen on dry ice. Testes were then surgically extracted and weighed to calculate gonadosomatic index (GSI=gonad mass/BM*100) for each subject (subject information available in Supplementary Data 1). Tissue blocks were stored in 5 ml CryoELITE tissue vials (Wheaton) at −80 °C until further processing.

Telencephala were cryo-sectioned coronally at 10 μm thickness using a Cryostar NX70 cryostat at −20 °C. Four tissue sections per subject were mounted onto 6.5 mm² capture areas on pre-cooled Visium Spatial Gene Expression slides (10x Genomics, 200233) and slides were stored at −80 °C for further processing. RNA quality of the tissue sections (RIN > 7) was confirmed using the Agilent RNA 6000 Nano Kit on the Bioanalyzer 2100 system (Agilent). Visium spatial gene expression slides were processed according to manufacturer instructions (10x Genomics; Methanol Fixation, H&E Staining, and Imaging – Visium Spatial Protocol CG000160). Slides were warmed to 37 °C for 1 min and fixed in methanol at −20 °C for 30 min, followed by isopropanol incubation for 1 min at room temperature. Tissue sections were then stained for hematoxylin and eosin (H&E). Brightfield images of H&E-stained slides were taken at 1.33 mm/pixel resolution using a Zeiss AxioObserver Z1 Fluorescent Microscope (Zeiss) with a 5X objective and stitched using Zen 2 software (blue edition, Zeiss) prior to library construction.

### Visium spatial gene expression library generation
Dual-index Illumina paired-end spatial gene expression libraries were prepared according to manufacturer instructions (10x Genomics; Visium Spatial Gene Expression Reagent Kits User Guide CG000239). Fixed and stained tissue sections were enzymatically permeabilized for 18 min. Optimal permeabilization time was determined to be 18 min based on initial tissue optimization trials (10x Genomics; Visium Spatial Tissue Optimization Reagent Kits User Guide CG000238). Poly-adenylated mRNA released from cells was captured by primers on the underlying spatially-barcoded 55 mm-diameter gene expression spots. Primers include a 10x spatial barcode, unique molecular identifier (UMI) and a partial read 1 sequencing primer (Illumina TruSeq Read 1). Incubation with reverse transcription (RT) reagents produced spatially barcoded, full-length cDNA. Barcoded cDNA was denatured, transferred into tubes, and amplified via PCR. Amplified cDNA was size-selected with SPRIselect. During final

library construction, P5 and P7 paired-end construct sequences, i5 and i7 sample indices, and a read 2 primer sequence (Illumina TruSeq Read 2) were added. Quality was assessed using high sensitivity DNA analysis on the Bioanalyzer 2100 system (Agilent). Libraries were pooled together and sequenced on the NovaSeq 6000 (Illumina) platform using the 150-cycle SP Reagent Kit (800 M reads; sample information available in Supplementary Data 1). The raw data was generated by ref. 15 and are publicly available in the National Center for Biotechnology Information (NCBI) Gene Expression Omnibus (GEO) under accession code GSE217615.

**Statistics and reproducibility**
Spatial transcriptomics data pre-processing and quality control. FASTQ files were processed with 10x Genomics Space Ranger 1.3.1. Reads were aligned to the *Maylandia zebra* Lake Malawi cichlid genome assembly[16] using a splice-aware alignment algorithm (STAR) within Space Ranger and gene annotations were obtained from the same assembly (NCBI RefSeq assembly accession: GCF_000238955.4, M_zebra_UMD2a). H&E images were oriented based on fiducial markers and the appropriate slide layout files were chosen. Space Ranger aligned barcoded spot patterns to the input slide image and distinguished tissue from background on the slide. Following these steps, Space Ranger generated filtered feature-barcode matrices (one per slide) containing expression data for a total of 32,471 features (corresponding to annotated genes) and a total of 6755 barcodes (corresponding to spots). In R the "Seurat" package was used to remove spots with less than one UMI and spots containing small pieces of stray tissue (48 spots removed).

Selection of tissue sections for downstream analysis. This analysis focuses on seven tissue sections from two subject males, S1 ($n = 3$ sections) and S2 ($n = 4$ sections). This decision was made based on (1) the high quality of the tissue sections, and (2) the observation that this combination of S1 and S2 sections yielded a comprehensive representation of telencephalic subregions along the rostral-caudal axis. The order of the tissue hemispheres along the rostral-caudal axis was determined through visual inspection of H&E-stained tissue images.

Clustering and selection of parameters. To prevent over-tuning of clustering parameters, a systemic method was used to determine optimal clustering parameters. ChooseR is a tool compatible with Seurat that evaluates clustering quality based on robustness metrics of bootstraps of the data. The code was adapted to evaluate more clustering parameters and to follow a slightly different workflow. The additional parameters that were evaluated were min.dist and n.neighbors in the RunUMAP function from Seurat. To ensure that bootstraps were entirely independent, the find_clusters function was modified to include the SCTransform, RunPCA, and RunUMAP functions from Seurat. Without these functions, there would be information leakage in the form of the variable genes and reduced dimensional space. In SCTransform, tissue sections were regressed out using var.to.regress and the resulting "SCT" assay was set as the active assay. 50 dimensions were used in RunPCA and RunUMAP. Additionally, in RunUMAP spread was set to 1, n.epochs was set to 1000 and metric was set to "euclidean". In FindNeighbors, reduction was set to "UMAP" and the first two dimensions were used, k.param was set to the same value as n.neighbors, n.trees was set to 500, and prune.SNN was set to 0. In FindClusters, algorithm was set to 2.

ChooseR was used to evaluate the clustering of combinations of min.dist and n.neighbors in RunUMAP and resolution in FindClusters. The set of values tested were as follows: min.dist of 0.1, 0.2, 0.3, 0.4, and 0.5; n.neighbors of 10, 20, 30, 40, and 50; and resolution of 0.2, 0.4, 0.6, 0.8, 1.0, and 1.2. The default value of 80% subsampling of the data was used as well as the default of 100 bootstraps (per combination of parameters). ChooseR defines the near-optimal clustering parameters as the one yielding the highest number of clusters whose median silhouette score is greater than the highest lower bound (95% confidence interval). For our ST data, this methodology identified the near-optimal clustering parameters to be

min.dist=0.1, n.neighbors=30, and Resolution=1.2, which we used to cluster spots.

Human Ortholog identification. Teleosts including *M.zebra* have undergone a genome duplication since their divergence with mammals (last common ancestor est. 360–450 MYA[36]), thus identification of human orthologs is non-trivial and multiple methods were used to ensure accuracy. Human orthologs to cichlid genes were retrieved from ENSEMBL and ref. 77, which uses a protein-based approach. To perform interspecies correlation analysis, one-to-one orthologs are required. For all orthologs, cichlid genes were kept only where the human ortholog was contained in one of the following: NCBI gene name, NCBI gene symbol, ENSEMBL gene symbol, ref. 77, and human ortholog from ENSEMBL. Due to the genome duplication, there are numerous many-to-one orthologs. In these cases, the cichlid gene with the greatest consistency among the aforementioned sources was kept. In cases of ties, the cichlid gene that contained the greatest number of UMIs in the spatial transcriptomics and single nuclei gene expression matrices was kept. All other cichlid genes were removed. This resulted in 13,237 cichlid genes for comparative analysis of shared marker genes. For interspecies integration analysis, a different method, BLAST, was used for ortholog identification (see 'Interspecies integration of single cell/nucleus RNA-seq data').

Computation of cell-type abundance estimation in spatial transcriptomics spots. For cell-type deconvolution of spots, cell2location[33] was used as it was among the top performing tools for this task reported by a recent benchmarking paper[34]. Cell2location is a Bayesian model that estimates the absolute abundance of cell-types at each spot. First, the regression model for the single cell data was initialized with default settings, using batch as the batch_key, and the model was trained using a maximum of 250 epochs. Next, the regression model for the spatial transcriptomics data was initialized with the single cell reference signatures, default settings, and hyperparameters selected based on cell2location's recommendations. We did not observe strong within-batch variation in total RNA count, therefore we set detection-n_alpha=200. By manual visual inspection of a subset of spots, we estimated the approximate mean number of cells per spot to be 8 and therefore set N_cells_per_location=8. The model was then trained using a maximum of 30,000 epochs. Finally, the estimated mean abundances of cell-types were rounded and spots with no estimated cells were assigned one cell to the cell-type with greatest abundance.

Manual cell abundance estimation in spatial transcriptomics spots. To evaluate the accuracy of the estimates of cell abundance in spatial spots by cell2location, we compared predicted cell numbers to our manual counts for a subset of spots in a single tissue hemisphere ($S_1C_2R$). Spots at the peripheral edges of the tissue contained a high density of cell bodies that could not be reliably distinguished and these spots were therefore excluded from the comparison. The Pearson's and Spearman's correlation coefficient was found between manual and computational estimates of cell abundance (Fig. S8).

scRNA-seq dataset retrieval. Mouse (est. divergence time from humans ~$112 \pm 3.5$ Myr[78]) gene expression data was retrieved from refs. 37,71. Gene expression data for other vertebrate taxa was retrieved from ref. 9 (axolotl, est. divergence time from humans ~$360 \pm 14.7$ Myr[78]), ref. 7 (turtle, est. divergence time from humans ~$222 \pm 52.5$ Myr[78]) and ref. 8 (songbirds, est. divergence time from humans ~$222 \pm 52.5$ Myr[78]). Next, the datasets were filtered to include only cells from the telencephalon. Data from ref. 37 was filtered to include only cells from cell-types annotated as belonging to the telencephalon and data from ref. 71 was only retrieved for the following brain regions: Frontal Cortex, Globus Pallidus, Hippocampus, Posterior Cortex, and Striatum. No filtering was required for the axolotl and turtle datasets, as the cells came from the telencephalon. All cells from the HVC and RA of

songbirds were kept, but only MSNs from Area X were retained due to non-descriptive annotations of the other cells. Next, cells annotated by ref. 37 as low-quality were removed. This information was not readily available for ref. 71, thus the dataset was subsetted to include only cells with greater than 500 UMIs, 500 genes, less than 5000 UMIs, and less than 2500 genes. The other vertebrate datasets retrieved were already filtered, so no further filtering was performed. Finally, due to the large number of cells from ref. 71, cells were randomly and evenly removed from the largest clusters until 100,000 cells remained. All datasets were normalized using SCTransform using default options.

**Evaluation of methods for comparative cell-type analysis.** The accuracy of two methods for comparison of cell-types across species were evaluated: correlation of common marker genes and SAMap (see Methods below). To determine their accuracy, they were tested by comparing cell-types whose relationships are already known. Thus, a scRNA-seq dataset from mouse telencephalon[37] was downsampled and compared to itself. Therein, the cell-types in the downsampled dataset are known to be the same as the original. To downsample the dataset, 10% of UMI's were removed from each cell and three cell-types were randomly chosen to be removed so the comparison was not one-to-one. Then the downsampled dataset and the original dataset were compared using the two methods. The resulting matrices of correlations/scores between cell-types in the downsampled and original dataset were compared to the identity matrix (i.e., same cell-types were set to 1 and others were set to 0). The coefficient of determination (squared Pearson's correlation) was found between the matrices resulting from the two methods and the identity matrix.

**Interspecies correlation of single cell/nucleus RNA-seq data.** To compare sc/snRNA-seq from the cichlid and mouse telencephalon, a procedure similar to refs. 7–9 was implemented as it is well suited for comparisons of distantly related species. Briefly, for each cell-type or brain region a Spearman correlation and corresponding $p$ value were calculated based on the average normalized expression of DEGs present in both species. For robustness, correlations were calculated using multiple mouse datasets[37,71]. Interspecies correlations using data from ref. 37 were performed on the "ClusterName" metadata and correlations using data from ref. 71 on the "subcluster" metadata.

DEGs (adjusted $p < 0.05$, log2FC > 0.25, and min.pct > 0.1) were found in the cichlid and mouse datasets using the FindAllMarkers functions with only.pos = T and otherwise default parameters. Then, the common set of genes that were DEGs in both species was found (see :Human Ortholog identification), and the average expression of these genes was calculated across the whole dataset using the AverageExpression function in Seurat. Similar to ref. 8, these average gene expression values were normalized by $\log(x + 1) + 0.1$ and were divided by the mean for each gene across the dataset. The normalized average gene expression matrices for the cichlid and mouse datasets were then correlated using a Spearman correlation. Significance of these correlations was determined using two criteria: (1) Bonferroni adjusted p-value of Spearman correlation test <0.05 and (2) Bonferonni adjusted $p$ value of permutation test <0.05. The former was calculated using cor.test in R with method = "spearman" and alternative = "greater". Similar to ref. 7, the latter was found by shuffling the normalized average gene expression values across cell-types in the cichlid gene by cluster matrix, then re-calculating the Spearman correlations with the mouse gene by cluster matrix. 1000 permutations were performed and the $p$ value was calculated by finding the proportion of correlations from the permutations greater than the real, non-permuted correlations.

**Interspecies integration of single cell/nucleus RNA-seq data.** Sc/snRNA-seq data from different vertebrate taxa were integrated to enable comparative analysis of transcriptional similarities of cell-types. To achieve this we used SAMap[13], a python package, as it was designed for comparative analyses of distantly related species. SAMap considers

sequence similarity of genes between species in its projection of the datasets in joint lower-dimensional spaces. This method allows genes with greater sequence similarity across species to weigh more heavily in the integration, which is advantageous when comparing species with large evolutionary distances like cichlids and mice. First, to determine sequence similarity, the proteomes of the vertebrate taxa were downloaded from NCBI using the version of the assemblies used to create the sc/snRNA-seq datasets: *Maylandia zebra* (GCF_000238955.4), *Mus musculus* (GCF_000001635.27), *Chrysemys picta bellii* (GCF_000241765.3) and *Taeniopygia guttata* (GCF_003957565.1). For the axolotl (*Ambystoma mexicanum*) dataset, the assembly was retrieved from v6.0 release of the genome not located on NCBI (AmexT_v47). Next, blastp was executed to identify reciprocal blast hits between vertebrate taxa and cichlids using SAMap's map_genes.sh script. Then, the raw gene by counts matrix of the sc/snRNA datasets from their Seurat objects were saved as an h5ad file and processed using the SAMAP function with arguments specifying the gene symbols of each protein and the directory containing the blastp results. Finally, the run function was executed using default parameters.

The knn-graph produced by SAMap was then used to create a similarity score between cell-types in cichlids and other vertebrates. The similarity score was defined as the mean number of k-nearest neighbors between cells from other vertebrates to cichlid nuclei. To determine significance, a permutation test was performed in which the cell-type labels of cells/nuclei for both datasets were shuffled and similarity scores were calculated. This was repeated 1000 times and cross-species cell-type pairs with similarity scores greater than all permutations of each cichlid cell-type were considered significant.

**Identification of genes driving cell-type relationships discovered using SAMap.** To find the genes driving cross-species cell-type relationships, the function GenePairFinder from SAMap was used with default parameters. For every pairwise combination of cell-types, the find_genes function from SAMap was used to find the genes driving the relationship, with n_genes set to the total number of genes in the datasets. The default parameter returns the top 1000 genes driving the relationship, but we allowed the function to return all the genes it found to drive the relationship. These genes are defined by SAMap as genes that are differentially expressed in each cell-type and contribute positively to the cross-species correlation of the cell-types.

**Identification of genes upregulated in conserved cell-types across vertebrates.** For strongly conserved cell-types, marker genes (significantly differentially expressed genes with adjusted $p < 0.05$, $\log_2$ fold change >0 and expressed in a minimum of 10% of cells/nuclei) with the same human ortholog in all datasets were found. Strongly conserved cell-types included: OLIG (mouse OEC; bird Oligo; turtle tsOlig; axolotl OLIG15; cichlid 2.2_Oligo), OB (mouse OBDOP2; bird GABA-1-1; turtle i01; axolotl GABA3; cichlid 5.2_GABA), MGE1 (mouse TEINH17; bird GABA-3; turtle i07; axolotl GABA2; cichlid 6_GABA), MGE2 (mouse TEINH21; bird GABA-2; turtle i08; axolotl GABA17; cichlid 15.3_GABA), MSN (mouse MSN1; bird MSN3; turtle i05; axolotl GABA11; cichlid 4.1_GABA), DGNBL (mouse DGNBL1; bird Pre-2; turtle tsNPCs; axolotl NB1; cichlid 9.5_Glut) and CA3 (mouse TEGLU23; bird HVC_Glut-3; turtle e34; axolotl GLUT7; cichlid 8.9_Glut). Additionally, human orthologs in conserved cell-types that were upregulated (positive $\log_2$FC value), but not meeting the other criteria of marker genes were found. When multiple genes in a species had the same human ortholog, the gene with the greatest $\log_2$FC was selected.

**Interspecies comparison of anatomical regions.** To gain insight into the similarity of anatomical regions across vertebrates, we compared scRNA-seq data from turtles and spatial transcriptomics data from mice to cichlid spatial transcriptomics data separately. Both analyses were performed in the same manner described in 'Interspecies integration of

single cell/nucleus RNA-seq data'. For the comparison to the turtle dataset, the detailed anatomical labels were used when available and the broad anatomical labels were used in all other cases. The mouse spatial transcriptomics dataset was retrieved from ref. 45 and the dataset was subset to include only cells from the cerebrum (Allen Brain Atlas abbreviation 'CH'), the ventricular systems (Allen Brain Atlas abbreviation 'VS'), and fiber tracts (Allen Brain Atlas abbreviation 'fiber tracts'). Then, the dataset was normalized using SCTransform, and the annotations from ABA were used (ABA_acronym column in the metadata). Since these annotations were extremely detailed, the parent annotation was used in some cases. Moreover, spots annotated as cortex ("CTX") with no more detail were assigned to "U_CTX" to represent undefined cortical regions.

**Composition of genes driving effects and conserved marker genes by transcription factors, neuromodulatory ligands and receptors.** The composition of genes driving relationships between cell-type pairs was investigated. The number of genes driving SAMap effects and the number of shared marker genes belonging to the following gene categories were found: transcription factors, neuromodulatory ligands, receptors, or "other". Transcription factors were obtained from ref. 7 and neuromodulatory ligands and receptors were obtained from ref. 15. The aforementioned lists are of human genes, thus for gene pairs identified by SAMap from non-human species, the human ortholog of the mouse gene was used when possible (i.e., in cichlid-mouse and turtle-mouse comparisons) and otherwise the human ortholog of cichlid genes was used (i.e., cichlid-turtle comparison). The number of genes from each category were found and compared for across cell-type hits. To determine significance, a $\chi^2$ test was performed on the mean number of transcription factors, neuromodulatory ligands, and receptors.

### Reporting summary
Further information on research design is available in the Nature Portfolio Reporting Summary linked to this article.

### Data availability
All data needed to evaluate the conclusions in this paper are present in the paper and/or the Supplementary Materials. The spatial transcriptomics data were generated by Johnson et al. 2023[15] and are publicly available in the National Center for Biotechnology Information (NCBI) Gene Expression Omnibus (GEO) under accession code GSE217615. Source data for all graphs presented in the main figures can be found in Supplementary Data 14.

### Code availability
Custom scripts for data analyses can be found at GitHub (https://github.com/ggruenhagen3/cichlid_st) and Zenodo (10.5281/zenodo.8364925).

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

## Acknowledgements

We dedicate this work to Karen Maruska for her groundbreaking research in teleost neuroanatomy and behavioral neuroscience. We thank the Georgia Tech Petit Institute Molecular Evolution Core for their integral roles in sequencing. This work was supported in part by NIH R01GM101095 and R01GM144560 to J.T.S., NIH F32GM128346 to Z.V.J. and Human Frontiers Science Program RGP0052/2019 to J.T.S., and OD P51OD011132 to Emory National Primate Research Center (Z.V.J.).

## Author contributions

B.E.H. and C.M.B. optimized spatial transcriptomics wetlab protocol. Z.V.J. and B.E.H collected samples and B.E.H. performed all downstream wetlab work and library preparation for spatial transcriptomics. The Petit Institute Molecular Evolution Core at GT performed sequencing for spatial transcriptomics. G.W.G. performed all bioinformatics and computational tasks, including: processing reads, quality control, clustering, marker gene analysis, prediction of the anatomical location of cell-types and cross-species gene expression comparisons. B.E.H. reviewed and synthesized literature on teleost neuroanatomy and cell-type biology and performed manual anatomical annotation of spatial transcriptomics spots. B.E.H. and G.W.G. reviewed literature on comparative vertebrate neuroanatomy. G.W.G. designed downstream statistical analyses with feedback from B.E.H., Z.V.J., and J.T.S., B.E.H., and G.W.G. took the lead on writing the manuscript with contributions from Z.V.J. and critical feedback from Z.V.J. and J.T.S., G.W.G. took the lead on designing and creating figures with critical feedback from B.E.H., Z.V.J., and J.T.S. J.T.S. funded spatial transcriptomics experiments.

## Competing interests

These authors declare no competing interests.
