## [Peer Review File · Communications Biology]

Reviewers' comments:

Reviewer #1 (Remarks to the Author):

Hegarty et al. performed spatial transcriptomics on the cichlid fish telencephalon and conducted a series of integrative analyses with single-nuclei RNA-seq datasets from the telencephalon of other vertebrates. They initially integrated the cichlid fish spatial transcriptomics dataset with its recently published single-nuclei RNA-seq dataset, allowing them to map cell types to their respective brain regions with the highest abundance. Subsequently, they conducted pairwise comparisons between regionally mapped cell types in the cichlid fish telencephalon and those of other vertebrate telencephala to identify homologous brain regions based on cell type conservation. They also directly compared their cichlid fish spatial transcriptomics data with spatially resolved turtle and mouse transcriptomics datasets to identify corresponding brain regions between cichlid fish and other vertebrates. This represents an important and ambitious goal, and the authors have, for the most part, employed appropriate tools and methodologies for their comparisons. Their spatial transcriptomics results remarkably capture the anatomical organization and appear to be a valuable resource for future studies. However, the presentation of the findings related to cross-species comparisons can be challenging to follow, and I have some reservations about whether certain interpretations are adequately supported.

Major Comments

1- I could not understand why the authors chose to perform pairwise integrations instead of integrating all species at once. This could help with anchoring the interpretation to mouse, the most well characterized organism among all species. The current representation makes it difficult to judge Figure 5 and Figure 6a (non-mouse comparisons). A single plot that shows the best mapping regions between mouse-axolotl-turtle-songbird-cichlid fish, along with an explanation of how these regions are selected, could help with a clear representation of the results.

2- The authors wrote:

“Based on our findings and previous literature, we propose the following brain regions contain molecularly conserved populations in cichlids, turtles, and mice: DI-g, aDVR/aDC, and neocortex; DI-vv, DMC, and CA3; Dm-2r, pDVR, and pallial amygdala; Dp, aLC, and piriform cortex.”

The authors base these conclusions on Figures 4 and 5 but looking at Figure 6a, it looks like Dp and Dm-2r are both similar to aLC and pDC. Also DI-vv is no more similar to DMC than aDC. Figure 6b also does not show significant similarity between Dm-2r and anything in mouse (including PA AMG). Figures 4 and 5 are based on cell type to region annotations that naturally have some error rate whereas Figure 6 is a region-region comparison and in theory should be a more direct comparison. The authors need to explain this discrepancy.

3- There seems to be some spurious cell type correspondence between neurons of fish and neurons of mouse (Figure 4c). It is hard to tell if the correspondences with $q_{perm}=0$ that are not highlighted by the authors should be taken seriously. Some of this could be because of an overall similarity between neurons compared to non-neurons, and between glutamatergic neurons compared to GABAergic neurons. This noise can be mitigated by subsetting the data per major cell type group and re-performing this

analysis. This is already done with GABAergic (Figure S12) and seems to yield cleaner results. It would be interesting to see both GABAergic and glutamatergic results with brain region annotations, and assess if Figures 4-5 and Figure 6 will correspond better.

Minor Comments

1- Consider specifying which marker gene(s) label which anatomical structures (Figure S4D)

2- Please show the hemisphere and individual breakdown of each cluster/region in a stacked barplot.

3- Criteria for marker gene selection does not have an effect size cutoff (\log_2 fold change > 0 . Seurat default is \log_2 FC > 0.25). This likely results in many weak marker genes.

4- SAMap versus marker gene correlation in Figure 3a is not very informative since any tool designed to integrate datasets will perform better than a crude analysis relying on correlating/overlapping cell type markers. Since benchmarking suggests it is among the best performing tools (<https://www.nature.com/articles/s41467-023-41855-w>), there is no need to include this comparison. Instead, it would be informative to show a breakdown of the UMAP in Figure 3b across the two species and their samples.

5- Cross species cell type mapping in Figure 4c seems like a proper statistical comparison and yields biologically meaningful results. But as with any cell type mapping method, the results may vary based on the accuracy and resolution of initial annotations. For example, the authors annotated oligodendrocytes as a single cell type whereas Ziesel et al divided them into NFOL, MFOL, MOL. I suspect that the nearest neighbors to 2.2_Oligo are shared among NFOL, MFOL, MOL which would also divide the correspondence strength. This may also affect the strength of correspondence for other cell types as well. I think it would be good to mention this confounding factor.

Reviewer #2 (Remarks to the Author):

A major unresolved issue in the field of brain evolution is the relationships among the brain regions of the major vertebrate clades. Over evolution, a wide range of neural structures have evolved across fish, amphibians, and amniotes, making it challenging to identify homologous brain regions across groups. A wealth of new analyses across a variety of organisms have characterized cell types and brain regions at the transcriptional level and have thus enabled high-resolution comparisons between species to better reconstruct evolutionary relationships.

A particularly thorny issue has been how the various structures of the teleost brain relate to those of other vertebrates. The teleost brain everts during development, while those of amniotes evaginate, resulting in particularly contrasting anatomy. In this study, Hegarty, Gruenhagen, Johnson, et al. provide a valuable addition to efforts to address this problem through an integration of spatial and cellular transcriptomics in the brain of a species of cichlid. The work leverages these datasets to perform interspecies comparisons with similar datasets obtained from the brains of axolotls, turtles, songbirds, and mammals. The authors find transcriptional similarities between teleost brain regions and defined

pallial domains in amniotes, suggesting deep conservation of basic vertebrate brain organization. In particular, they find evidence that certain regions of the teleost brain show regional similarity to the dorsal pallium, suggesting that the regions and cell types that typify the mammalian neocortex were present in the last common ancestor teleosts and tetrapods. The data is of high quality; the analysis is thorough, rigorous, and beautifully presented; and the authors' interpretations and conclusions are largely supported by their findings. This dataset and the extensive annotation and interpretation provided here will serve as an excellent resource for the field. Along with several minor comments, I have a few major comments that I believe need to be addressed.

Major

Lines 447-460: What is the rationale for this analysis comparing the representations of different gene classes within the driving-gene sets? If it is to argue that, in contrast to past work in turtles and songbirds that found differences in regional associations when using subclasses of genes, in fish no such pattern is found, then this analysis is not persuasive. The same proportion of gene classes could be found across comparisons yet the individual genes within each of the classes could differ. It would be informative to perform a similar analysis as used in Tosches 2018 and Colquitt 2021 in which cell type similarities are assessed using gene subclasses.

Figure 5C: It appears that the strongest similarities of the DI-g neuron types are to cingulate/RSP types. The cingulate has a different structure from the neocortex proper and is classified as "mesocortex" by some authors (Puelles, L., Alonso, A., García-Calero, E. & Martínez-de-la-Torre, M. Concentric ring topology of mammalian cortical sectors and relevance for patterning studies. *J. Comp. Neurol.* 527, 1731–1752 (2019)) and as part of the "dorsomedial" pallium by others (Pattabiraman, K. et al. Transcriptional Regulation of Enhancers Active in Protodomains of the Developing Cerebral Cortex. *Neuron* 82, 989–1003 (2014)). This framing would fit well the position of DI-g relative to the medial pallium. You describe these populations as neocortical (last paragraph of the Discussion), but some discussion of the distinct classification of cingulate/RSP neurons would strengthen your analysis.

Figure 7B: Turtle e02 also shows significant similarity to mouse TEGLUT17 (piriform), according to Fig 7A and Fig. S15. Why was this comparison not included in Figure 7B? That is, why were only neocortical-like clusters included? Similarly, several aDC types show strong similarity to several non-dorsal pallial mouse types, including TEGLUT13/14 (subiculum), 15/16/17 (piriform), and 18/19 (anterior olfactory nucleus). These similarities indicate that, in this analysis, there is not a clean association between aDC and DP. Some description of and reasoning through this ambiguity should be provided.

Lines 571-598: The transcriptional similarities between glutamatergic neurons in aDVR/aDC, DI-g, and some dorsal pallial regions are compelling. However, the aDVR is fairly clearly not a dorsal pallial field homolog, so the authors should take care in separating cell type versus field homology. I am convinced that DI-g is a good candidate for dorsal pallial homology, but less convinced that these results require a "remapping" of aDVR as a neocortical/dorsal pallial homolog. Glut types in aDVR and HVC in songbirds (contained within the nidopallium, the field homolog of the aDVR) are similar to neocortical types when comparing across all genes but appear more similar to non-dorsal pallial domains when using transcription factors, suggesting that there is some separation between regional and functional identity.

Minor

Line 71: In the methods or supplement, please provide some measure of similarity between the species used in this study and the reference genome species, e.g. time since divergence.

Line 76: Here and in subsequent usages, the description of the clusters as “unbiased” seems unwarranted. “Data-driven” or since this clustering was obtained using standard approaches, simply “clusters” would suffice.

Lines 234-240: You transition to describing MGE-class type neurons but this isn’t obvious. Please indicate that these classes are similar to MGE neurons.

Line 261: The p-values for conserved marker genes and SAMap driving gene differences are different from those given in the legend for Figure 4D and E. Please resolve.

Line 295: It looks like you’re including data from songbird Area X as well. Please include this in your description of the datasets in the Results.

Lines 330-33: 9.8_Glut also shows similarity to RA_Glut, a pDVR/arcopallium neuron type in songbirds.

Lines 582-584: “Additionally, previous analyses found that songbird cell types most similar to the turtle aDVR (HVC_Glut-2 and HVC_Glut-5) also resembled the mammalian neocortex using both effector genes and transcription factors separately”. I don’t think this is true. This study did find HVC_Glut-2 and Glut-5 non-TF similarities to the cortex but largely found TF similarities to ventral pallium structures.

Figure 4A: What is the x-axis here? The full Zeisel dataset? If so, please make this more explicit.

Figure 4C: Please define RHP.

Figure 7 legend: “Full results” seems to correspond to Fig. S15 and not Fig. S14 as written.

Reviewer #3 (Remarks to the Author):

Hegarty et al. present a spatial transcriptomics dataset obtained from adult Mchenga conophoros telencephalon, a bower-building cichlid. This study is a follow-up of a previous snRNAseq study of the same animal, in which they identified cell types that might be specifically associated with the peculiar behavior of this species. In the current study, the authors perform a comprehensive analysis of the spatial transcriptome of male adult telencephalon and link the data to their previous snRNAseq dataset to delineate and annotate anatomical regions in the adult telencephalon. Next they use SAMap to compare the previously collected telencephalon snRNAseq dataset with four additional vertebrates

representing the different clades (amphibian, reptile, bird, mammal). This analysis yields highly interesting findings regarding conservation of cell types across vertebrate evolution.

Major strengths of the paper are the unique spatial dataset that was collected, the careful analysis of the spatial dataset and anatomical mapping, the extensive comparison with other vertebrate telencephalon datasets to get a comprehensive evolutionary picture. Overall, methods used are sharp, and careful statistical analyses has been performed. Quality tests for snRNA-seq datasets were done comparably across the datasets used.

A weakness is the absence of comparison with other teleost models like zebrafish (see details below). A final scheme including all species compared at adult stage would be informative. The comparison with the mouse brain in Figure7e has issues that I have detailed below.

Major comments

An important aspect of the study is the comparison of their single-nuclei dataset with other species datasets. Here the authors mention they used SAMap to calculate a similarity score that was compared to permutations, and that they show those that have a score that is greater than all permutations. This leads to different scales in figure 4 and 5 when comparing to mouse, axolotl, turtle or songbird. Can they explain why these values are different? In addition, as each scale is different, the reader cannot distill really to what extent the comparisons remain meaningful. Is a mid-scale hit meaningful or not? Do the authors wish to say that darkest color hits suggest true homology, and in the absence of that, a cell type might be “novel”? But since we cannot really compare between species, the cutoff of a meaningful hit might be different on different comparisons? It would be nice to better understand the view of the authors here. In view of evolution, the comparison with all these species is a great resource!

I missed the comparison with zebrafish, especially given the fact the authors suggest an evolutionary comparative picture that goes against the current view on zebrafish telencephalon organisation. Does the cichlid have a different telencephalic build-up or is the current zebrafish map incorrect? What is known from other teleost brains like Medaka or goldfish? Including this comparison would also give the authors the opportunity to discover novelty. At the moment, it would be unclear if non-matching or less-matching cell types are cichlid specific or teleost specific.

The scheme in Figure7e compares “adult” mouse with cichlid telencephalon. Although the mouse section was taken from a previously published paper, it depicts an embryonic mouse brain, not an adult brain. The Allen brain atlas provides suitable coronal sections for comparison of brain regions in the adult mouse. The issue with Figure7e is that here, the authors in fact depict embryonic origin of regions MP, DP, LP, VP, subpallium and display it as if these were fixed areas in the adult brain as well. The issue is that there is a lot of cell migration and cells from VP and subpallium end up in other adult areas, making the adult picture more complex. The question is to what extent the study, performed on adult specimen, can really disentangle this complexity, given the limitations of resolution of the visium and despite the powerful analysis the authors have done by implementing their single-nuclei dataset in the spatial data (figure 3). It would be really helpful if the authors could clarify what assumptions they can or want to make.

Minor comments

Reads were mapped to the Maylandia zebra reference genome- to what extent is this reference genome similar to the Mchenga conophoros genome? Can the authors give numbers of similarity on the transcriptomic sequence level?

The authors identify cell types that seem to be GLUT and GABA: is this a unique teleost cell type? can they explain?

Line 200: wrong figure panel cited?

Lines 580-2: In contrast, our SAMap integration revealed no significant difference in the number of transcription factors (or other gene categories) driving relationships between cell-types in the mouse neocortex and turtle aDVR/aDC.

The data sampled here are taken from adult fish, that might have already downregulated TF expression. To what extent might this affect the authors' conclusions?

Lines 593-595 Taken together, our results and those from previous analyses[8,9] suggest that conserved neuronal cell-types in the teleost forebrain may represent precursors to well-studied neocortical populations in the mammalian brain.

The term "precursor" suggests these cell types might have the ability to mature or give rise to mammalian neocortical cell types. In view of how evolution might have worked, this is rather unlikely. Would it not be more likely that fish cell types x y z might be more similar to one or more putative ancestral cell types that gave rise to an expanded array of cell types in the mammalian neocortex? Can the authors rephrase this?

This study is largely a meta-analysis using SAMap and no other tool was used to validate key findings. The authors should address this in their discussion.

Figure 4A Zeisel et al-

Figure 4 similarity scores panel: axis cell types names are barely readable; this is an issue with other figures as well; change size of those panels (if possible)?

General Response: Major revisions have been made following excellent and thought-provoking feedback
from reviewers. The main points of revision are as follows, 1) highlighted key signals from matches that
are many-to-one by noting reciprocal top hits, 2) performed a new major comparative analysis with a
recently-published atlas of the goldfish brain, 3) reinterpreted our results concerning the turtle pallium
given the direction of evidence in the field and the opinion of the reviewers, 4) modified the discussion of
vertebrate forebrain evolution with a more nuanced stance of the relation of cortical structures, including
the mesocortex. These revisions resulted in significant changes to the text and figures. Changes to the text
have been highlighted and are described in the responses below. Here are the modified figures w/ the
changes described:

**Figure 4.** Reciprocal top hits denoted with crosses in panel c.

Figure 5. Reciprocal top hits denoted with crosses in panels a-c.

**Figure 6.** Reciprocal top hits denoted with crosses in panels a-b.

**Figure 7.** Text in panel b for the mouse region row was changed from “neocortex” to “cortex”.

**Figure S11.** Reciprocal top hits denoted with crosses.

**Figure 12.** A new figure was added visualizing the results from a comparative analysis of the cichlid
telencephalon to the recently published goldfish telencephalon.

Reviewer #1 (Remarks to the Author):

Hegarty et al. performed spatial transcriptomics on the cichlid fish telencephalon and conducted
a series of integrative analyses with single-nuclei RNA-seq datasets from the telencephalon of
other vertebrates. They initially integrated the cichlid fish spatial transcriptomics dataset with its
recently published single-nuclei RNA-seq dataset, allowing them to map cell types to their
respective brain regions with the highest abundance. Subsequently, they conducted pairwise
comparisons between regionally mapped cell types in the cichlid fish telencephalon and those of
other vertebrate telencephala to identify homologous brain regions based on cell type
conservation. They also directly compared their cichlid fish spatial transcriptomics data with
spatially resolved turtle and mouse transcriptomics datasets to identify corresponding brain
regions between cichlid fish and other vertebrates. This represents an important and ambitious
goal, and the authors have, for the most part, employed appropriate tools and methodologies for
their comparisons. Their spatial transcriptomics results remarkably capture the anatomical
organization and appear to be a valuable resource for future studies. However, the presentation
of the findings related to cross-species comparisons can be challenging to follow, and I have
some reservations about whether certain interpretations are adequately supported.

Major Comments

1- I could not understand why the authors chose to perform pairwise integrations instead of
integrating all species at once. This could help with anchoring the interpretation to mouse, the
most well characterized organism among all species. The current representation makes it
difficult to judge Figure 5 and Figure 6a (non-mouse comparisons). A single plot that shows the
best mapping regions between mouse-axolotl-turtle-songbird-cichlid fish, along with an
explanation of how these regions are selected, could help with a clear representation of the
results.

Response: We agree with this comment, and in fact initially tried this approach before settling on pairwise
comparisons. Our conclusion was that the current available integration methods do not appear to be
sufficient to accomplish simultaneous integration of five species, all with vast evolutionary distances
between each other. As an example, our integration of all species resulted in seemingly poor performance,
with cell-types that have previously demonstrated strong similarity such as non-neuronal cell-types
(Tosches et al 2018 and Colquitt et al 2021), showing weak mapping. In lieu of a method that can
accurately accommodate such an integration, we have chosen to perform pairwise integration and start
with a comparison to mouse cell-types to ground the reader with this well-characterized species. This
pairwise approach resulted in cleaner mappings of core non-neuronal “ground truth” cell types, and has
been used effectively in comparisons of marine larvae
(<https://www.science.org/doi/10.1126/sciadv.adg6034>). Furthermore, we feel that visualizing the
comparisons in a pairwise manner may be easier to interpret than a 5-dimensional heatmap. To further aid
in interpretability across comparisons we have maintained the same order for the cichlid cell-types.

2- The authors wrote:

“Based on our findings and previous literature, we propose the following brain regions contain
molecularly conserved populations in cichlids, turtles, and mice: DI-g, aDVR/aDC, and
neocortex; DI-vv, DMC, and CA3; Dm-2r, pDVR, and pallial amygdala; Dp, aLC, and piriform

cortex.”

The authors base these conclusions on Figures 4 and 5 but looking at Figure 6a, it looks like Dp
and Dm-2r are both similar to aLC and pDC. Also DI-vv is no more similar to DMC than aDC.
Figure 6b also does not show significant similarity between Dm-2r and anything in mouse
(including PA AMG). Figures 4 and 5 are based on cell type to region annotations that naturally
have some error rate whereas Figure 6 is a region-region comparison and in theory should be a
more direct comparison. The authors need to explain this discrepancy.

Response: We did indeed base these conclusions primarily on Figures 4 and 5, with general agreement
found in Figures 6, S9, S11, S12, S16 (for example see zoom-ins of these figures below for cichlid DI-vv,
DMC, and CA3). We find that DI-vv does show greater similarity to DMC than aDC (13% increase).
Additionally, Dp and Dm-2r do show similarity to aLC and pDC, however they both show much greater
similarity to their stated regions, pDVR and aLC respectively. In both cases, they are reciprocal top hits
with each other. Cell-type comparisons tend to not result in exact one-to-one results as the algorithms are
designed to find cell-type similarities. As an example, glutamatergic neurons have similar features to one
another, and thus comparative analysis reveals similarities between glutamatergic neurons across species.
The greatest similarity of cell-types is of greater interest however as it reveals, which of the glutamatergic
neurons are the most similar. Thus, we place the greatest emphasis on cell-type and region relationships
that are reciprocal top hits and that are generally consistent across multiple analyses. In the main text, we
attempted to convey this mainly at Lines 198 and 381-383, with other references throughout. Region-
region comparisons are utilized because they are more direct, however the results may be influenced by
confounding factors such as composition of cell-types within region. By focusing on results that are
generally consistent across multiple analyses, we attempt to avoid weaknesses of any single approach. For
greater clarity, we have added visual cues to indicate reciprocal top hits, have included potential weakness
focusing on just cell-type or region analysis, and have added more statements in the text that inform the
reader that we report results that are consistent across analyses (Lines 302, 314, 326, 381-383).

3- There seems to be some spurious cell type correspondence between neurons of fish and
 neurons of mouse (Figure 4c). It is hard to tell if the correspondences with $q_{perm}=0$ that are not
 highlighted by the authors should be taken seriously. Some of this could be because of an
 overall similarity between neurons compared to non-neurons, and between glutamatergic
 neurons compared to GABAergic neurons. This noise can be mitigated by subsetting the data per
 major cell type group and re-performing this analysis. This is already done with GABAergic
 (Figure S12) and seems to yield cleaner results. It would be interesting to see both GABAergic
 and glutamatergic results with brain region annotations, and assess if Figures 4-5 and Figure 6
 will correspond better.

Response: In some circumstances, it can be difficult to determine if some hits are spurious or not,
 especially as the relationship between cell populations in the fish brain and other tetrapods is unclear.
 Additionally, we note that uncertainty can be introduced in cross-species mapping due to differences in
 dissections, clustering, and cell type/anatomical region annotations established by the authors (we have
 added a comment in our discussion section that addresses this topic (Lines 520-531)). In this manuscript,

we choose to highlight relationships supported by multiple analyses and previous studies. For example,
the relationship between the Dl-v and the hippocampus is supported by comparisons to two mouse
datasets, comparisons to other hippocampal-like populations in tetrapods, spatial transcriptomic analysis
and previous studies including lesion studies in goldfish. We also note that in some cases, ‘spurious’ cell-
type correspondence is likely due to similar gene expression profiles in cell-types that may be located in
different anatomical regions across species. For example, in Fig. 4C, cichlid 8.7_Glut shows strong
transcriptional similarity with mouse OBNBL1, annotated as glutamatergic neuroblasts from the olfactory
bulb. 8.7_Glut spatially maps to the cichlid Dp region, which is primarily glutamatergic and is compared
to the mammalian olfactory/piriform cortex, in part because it is the main recipient of olfactory input in
the teleost telencephalon. Unlike the restricted niches of adult neurogenesis in the mammalian brain, and
due to its strong connections with the olfactory bulb, it is not surprising that cells within the Dp region
may exhibit transcriptional similarity to mammalian olfactory neuroblasts. Moreover, we feel that more
confidence may be placed in mappings that include all cell-type classes and result in major cell-type
classes mapping correctly. Whereas, if mapping were to be performed within cell-type class and the
ground truth is not known, it may be more difficult to determine if the method worked well. Cell-types
mapping to the correct major cell-type class can provide a good sanity check of the results. Figure S12
(now Figure S13) shows results from analyses conducted with all cells, but the figure only visualizes the
GABAergic cell-types.

Minor Comments

1- Consider specifying which marker gene(s) label which anatomical structures (Figure S4D)

Response: In this supplementary figure, we chose to report the top markers for our unbiased spatial
cluster data without introducing anatomical identity, which was manually assigned for spots and outlined
in subsequent figures. In Figure S5 B, the composition of unbiased spatial clusters within regions is
shown, and Figure S6 contains marker genes for our data grouped by manually assigned anatomical
identities. We feel that introducing anatomical structure information into Figure S4, prior to introducing
how anatomical labels were assigned, may be misleading. To address this comment, in the main text
where Fig. S4 is called out (Line 79), we have additionally referenced the Supplementary Results, which
outlines the expression information of many marker genes in Figure S4D that have been reported on in
teleost literature and helped inform our anatomical region annotations.

2- Please show the hemisphere and individual breakdown of each cluster/region in a stacked
barplot.

Response: The number of spots from each unbiased cluster in each hemisphere can be found in Fig S4
(panel b). The correspondence between unbiased clusters and regions can be found in S5 (panel b) along
with hemispheres colored by regions to quickly assess which regions are in which hemispheres visually
(panel a).

3- Criteria for marker gene selection does not have an effect size cutoff (\log_2 fold change > 0 .
Seurat default is \log_2 FC > 0.25). This likely results in many weak marker genes.

Response: Good catch, this was not described adequately in the methods. The default values for the
FindAllMarkers function in Seurat were used, which are \log_2 FC > 0.25 and min.pct > 0.1 . We have
clarified this point in the methods (Line 855).

4- SAMap versus marker gene correlation in Figure 3a is not very informative since any tool
designed to integrate datasets will perform better than a crude analysis relying on
correlating/overlapping cell type markers. Since benchmarking suggests it is among the best
performing tools (<https://www.nature.com/articles/s41467-023-41855-w>), there is no need to
include this comparison. Instead, it would be informative to show a breakdown of the UMAP in
Figure 3b across the two species and their samples.

Response: The recent benchmarking paper is an excellent resource for future comparative analysis of
single cell RNA-seq data. Due to the recency of the benchmarking (Oct 14, 2023) and the prevalence of
correlating marker genes, we feel that this is an important point to make. High-impact studies published in
2022 (<https://www.science.org/doi/10.1126/science.abp9262>) and even in 2023 (albeit for more closely
related species: <https://www.nature.com/articles/s41559-023-02238-y>), continue to use the correlation of
marker genes as a tool for comparative analysis. Moreover, given the historic context for this method in
pioneering papers by Tosches et al in 2018 and Colquitt et al in 2021, we feel that noting the advantages
of this method will help sway future studies towards more comprehensive tools. Our analysis also
corroborates the benchmarking paper cited by the reviewer and can serve to boost the confidence of its
findings. Finally, we hope that it will boost the confidence of readers in our findings.

5- Cross species cell type mapping in Figure 4c seems like a proper statistical comparison and
yields biologically meaningful results. But as with any cell type mapping method, the results may
vary based on the accuracy and resolution of initial annotations. For example, the authors
annotated oligodendrocytes as a single cell type whereas Ziesel et al divided them into NFOL,
MFOL, MOL. I suspect that the nearest neighbors to 2.2_Oligo are shared among NFOL,
MFOL, MOL which would also divide the correspondence strength. This may also affect the
strength of correspondence for other cell types as well. I think it would be good to mention this
confounding factor.

Response: We agree that the way cell-types are divided and the level of specificity may affect the results.
This is now mentioned as a confounding factor at Lines 520-522. However, since the similarity score is
based on the mean rather than total number of knn neighbors, this prevents excessive sensitivity to the
number of divisions. For example, if a cell-type was divided evenly into multiple sub-cell-types and so
were its nearest neighbors then the similarity scores for the sub-cell-types would be unchanged.

Reviewer #2 (Remarks to the Author):

A major unresolved issue in the field of brain evolution is the relationships among the brain
regions of the major vertebrate clades. Over evolution, a wide range of neural structures have
evolved across fish, amphibians, and amniotes, making it challenging to identify homologous
brain regions across groups. A wealth of new analyses across a variety of organisms have
characterized cell types and brain regions at the transcriptional level and have thus enabled
high-resolution comparisons between species to better reconstruct evolutionary relationships.

A particularly thorny issue has been how the various structures of the teleost brain relate to
those of other vertebrates. The teleost brain everts during development, while those of amniotes
evaginate, resulting in particularly contrasting anatomy. In this study, Hegarty, Gruenhagen,
Johnson, et al. provide a valuable addition to efforts to address this problem through an

integration of spatial and cellular transcriptomics in the brain of a species of cichlid. The work
leverages these datasets to perform interspecies comparisons with similar datasets obtained
from the brains of axolotls, turtles, songbirds, and mammals. The authors find transcriptional
similarities between teleost brain regions and defined pallial domains in amniotes, suggesting
deep conservation of basic vertebrate brain organization. In particular, they find evidence that
certain regions of the teleost brain show regional similarity to the dorsal pallium, suggesting that
the regions and cell types that typify the mammalian neocortex were present in the last common
ancestor teleosts and tetrapods. The data is of high quality; the analysis is thorough, rigorous,
and beautifully presented; and the authors' interpretations and conclusions are largely
supported by their findings. This dataset and the extensive annotation and interpretation
provided here will serve as an excellent resource for the field. Along with several minor
comments, I have a few major comments that I believe need to be addressed.

Major

Lines 447-460: What is the rationale for this analysis comparing the representations of different
gene classes within the driving-gene sets? If it is to argue that, in contrast to past work in turtles
and songbirds that found differences in regional associations when using subclasses of genes,
in fish no such pattern is found, then this analysis is not persuasive. The same proportion of
gene classes could be found across comparisons yet the individual genes within each of the
classes could differ. It would be informative to perform a similar analysis as used in Tosches
2018 and Colquitt 2021 in which cell type similarities are assessed using gene subclasses.

Response: Thank you for the feedback, we have modified this section of text (Lines 448-490) to more
concisely and accurately explain our rationale. In previous studies, authors have subset the types of genes
included in analyses because they hypothesized that certain gene sets (TFs) might better reflect
evolutionary history than other gene sets. This is a reasonable hypothesis but has not been explicitly
studied to our knowledge. When subsetting the data, previous groups have found different homology
assignments (e.g., in Tosches et al 2018 TF analysis finds that the turtle pallial thickening is significantly
similar to the claustrum, while all analysis of all genes finds it's significantly similar to the neocortex).
These comparisons differ in gene type but also in the number of genes considered. Our rationale was to
use an unbiased integration approach (SAMap) using tens of thousands of genes and then to query
specific homology assignments to see if the genes driving integration differed. For instance, we might
have found that relationships among regions potentially similar to the mammalian neocortex were driven
predominantly by TFs while others were driven by receptors or ligands. However, that is not what we
found, instead there was no difference in the contribution of different gene types. Note that for these tests
we used the same gene lists as in Tosches et al 2018 for TFs and Johnson et al 2023 for other classes. For
241 us, it does not invalidate the previous analysis, but it does suggest that there is no reason in our case to
242 prioritize one interpretation over another based on gene class sub-setting.

Figure 5C: It appears that the strongest similarities of the DI-g neuron types are to
cingulate/RSP types. The cingulate has a different structure from the neocortex proper and is
classified as "mesocortex" by some authors (Puelles, L., Alonso, A., García-Calero, E. &
Martínez-de-la-Torre, M. Concentric ring topology of mammalian cortical sectors and relevance
for patterning studies. J. Comp. Neurol. 527, 1731–1752 (2019)) and as part of the
"dorsomedial" pallium by others (Pattabiraman, K. et al. Transcriptional Regulation of Enhancers

Active in Protodomains of the Developing Cerebral Cortex. Neuron 82, 989–1003 (2014)). This
framing would fit well the position of DI-g relative to the medial pallium. You describe these
populations as neocortical (last paragraph of the Discussion), but some discussion of the
distinct classification of cingulate/RSP neurons would strengthen your analysis.

Response: We are glad that this point was brought up, as we have done further research into the RSP-
cingulate cortex and its role in spatial behavior since submission to Communications Biology because we
consider the transcriptional similarity between cell-types in these regions and those in the DI-g a very
interesting result. To address this comment, we have added text in the discussion that delves a bit deeper
into how this region is anatomically positioned in the mammalian brain (in particular, in the mouse brain)
and have referenced the two publications listed in the reviewer’s comment to illustrate that it is classified
differently in the literature. We highlight the position of the RSP relative to the hippocampus and the
position of the DI-g relative to the DI-v as another layer of similarity prior to commenting on the roles of
these regions in spatial and social memory and behavior. We also note that the mesocortex and the
neocortex develop from the dorsal pallium in our results, lines: 452 and 599-601, and cite Puelles et al.
2019, to prime readers for the subsequent call-out to the mesocortex in our discussion.

Figure 7B: Turtle e02 also shows significant similarity to mouse TEGLUT17 (piriform), according
to Fig 7A and Fig. S15. Why was this comparison not included in Figure 7B? That is, why were
only neocortical-like clusters included? Similarly, several aDC types show strong similarity to
several non-dorsal pallial mouse types, including TEGLUT13/14 (subiculum), 15/16/17
(piriform), and 18/19 (anterior olfactory nucleus). These similarities indicate that, in this analysis,
there is not a clean association between aDC and DP. Some description of and reasoning
through this ambiguity should be provided.

Response: Thank you for the feedback, we made significant modifications to this section to clarify our
rationale (Lines 448-476). Our rationale for investigating the relationship between the aDC and aDVR to
the neocortex is that both structures have been compared to it in the literature. Furthermore, we find
significant similarities between cichlid DI-g cell-types to both the aDC and aDVR cell-types and between
cichlid DI-g cell-types and the neocortex. Thus, we sought to better understand the relationship between
these structures by using a newer approach, SAMap. The reviewer is correct that there are significant
similarities between these turtle cell-types and non-neocortical cell-types. However, here and throughout
the paper, we focus on relationships supported by or proposed in previous literature and that are
reciprocal top hits. In the main text, we attempted to convey this mainly at lines 198 and 381-383, with
other references throughout.

Lines 571-598: The transcriptional similarities between glutamatergic neurons in aDVR/aDC, DI-
285 g, and some dorsal pallial regions are compelling. However, the aDVR is fairly clearly not a
286 dorsal pallial field homolog, so the authors should take care in separating cell type versus field
homology. I am convinced that DI-g is a good candidate for dorsal pallial homology, but less
convinced that these results require a “remapping” of aDVR as a neocortical/dorsal pallial
homolog. Glut types in aDVR and HVC in songbirds (contained within the nidopallium, the field
homolog of the aDVR) are similar to neocortical types when comparing across all genes but
appear more similar to non-dorsal pallial domains when using transcription factors, suggesting
that there is some separation between regional and functional identity.

Response: We agree that these results do not necessitate a "remapping" of the aDVR. We find similarities
between the DI-g and the aDC and aDVR. Given the debate over the homology between the aDVR and
aDC to the neocortex, we sought to make the comparison using a new tool. We agree with this comment
and note that the aDC is increasingly considered to share evolutionary origins with the mammalian cortex,
thus we have re-interpreted our results, giving more reasoning through these relationships, Lines 483-489
and 627-649. Overall, our results are consistent with previous work demonstrating some transcriptional
similarities between both the aDC and aDVR with the mammalian cortex.

Minor

Line 71: In the methods or supplement, please provide some measure of similarity between the
species used in this study and the reference genome species, e.g. time since divergence.

Response: Thank you for this suggestion! We have added estimated time since divergence (from: Kumar,
S. & Hedges, S. B. A molecular timescale for vertebrate evolution. *Nature* **392**, 917-920 (1998)) for the
vertebrate groups used in our comparative analyses in the method section "scRNA-seq dataset retrieval".

Line 76: Here and in subsequent usages, the description of the clusters as "unbiased" seems
unwarranted. "Data-driven" or since this clustering was obtained using standard approaches,
simply "clusters" would suffice.

Response: We appreciate the suggestion and have removed the word "unbiased" for concision and clarity.
However, we note that we performed a unique approach to ensure that our clustering was near-optimal.
Perhaps the most common and standard approach used for determining clusters is to qualitatively evaluate
the clusters by their appearance in UMAP space. In contrast, our approach sought to determine clustering
parameters, and thus clusters, in an unbiased and quantitative manner using a computational tool called
ChooseR.

Lines 234-240: You transition to describing MGE-class type neurons but this isn't obvious.
Please indicate that these classes are similar to MGE neurons.

Response: Thank you for pointing this out, we have added this information to this sentence to indicate a
transition to our discussion of MGE-class interneurons.

Line 261: The p-values for conserved marker genes and SAMap driving gene differences are
different from those given in the legend for Figure 4D and E. Please resolve.

Response: Thank you for catching this, we have fixed this!

Line 295: It looks like you're including data from songbird Area X as well. Please include this in
your description of the datasets in the Results.

Response: Thank you for catching this, we have added Area X into this sentence.

Lines 330-33: 9.8_Glut also shows similarity to RA_Glut, a pDVR/arcopallium neuron type in
songbirds.

Response: Thank you for bringing this to our attention, however we believe you may be referring to 8-
9_Glut rather than 9.8_Glut. In Fig. 5C, 8-9_Glut is the cichlid cell-type which shows similarity to
RA_Glut songbird cell-types.

Lines 582-584: “Additionally, previous analyses found that songbird cell types most similar to
the turtle aDVR (HVC_Glut-2 and HVC_Glut-5) also resembled the mammalian neocortex using
both effector genes and transcription factors separately”. I don’t think this is true. This study did
find HVC_Glut-2 and Glut-5 non-TF similarities to the cortex but largely found TF similarities to
ventral pallium structures.

Response: Previous analyses of songbird cell-type did generally find more similarities between
glutamatergic neurons and ventral pallial structures when using only transcription factors (Colquitt et al
2021). However, when attention is paid particularly to HVC_Glut-2 and Glut-5, interesting patterns are
observed when considered in the context of our analyses. First, Figure 3F from Colquitt et al 2021 shows
that songbird HVC_Glut-2 and Glut-5 were the most similar to the turtle aDVR (further supported by
S5B). Figure S5B shows that HVC_Glut-2 and Glut-5 were most similar to mouse neocortical cell-types
when analyzed by both effector genes and transcription factors separately. In the TF analysis, HVC_Glut-
2 was most similar to TEGLU6 from cingulate layer 2 and HVC_Glut-5 was most similar to TEGLU8
from cortex layer 4 (with no significant similarities to any other cell-type).

Figure 4A: What is the x-axis here? The full Zeisel dataset? If so, please make this more
explicit.

Response: On the x-axis are all the telencephalic cell-types present in the Zeisel dataset. We have added
more details to the figure legend for clarity.

Figure 4C: Please define RHP.

Response: Thank you for pointing this out. We have addressed this by spelling out ‘Retrohippocampal
region’ on the figure for clarity.

Figure 7 legend: “Full results” seems to correspond to Fig. S15 and not Fig. S14 as written.

Response: Thank you for catching this. Fixed!

Reviewer #3 (Remarks to the Author):

Hegarty et al. present a spatial transcriptomics dataset obtained from adult Mchenga
conophoros telencephalon, a bower-building cichlid. This study is a follow-up of a previous
snRNAseq study of the same animal, in which they identified cell types that might be specifically
associated with the peculiar behavior of this species. In the current study, the authors perform a
comprehensive analysis of the spatial transcriptome of male adult telencephalon and link the
data to their previous snRNAseq dataset to delineate and annotate anatomical regions in the
adult telencephalon. Next they use SAMap to compare the previously collected telencephalon
snRNAseq dataset with four additional vertebrates representing the different clades (amphibian,

reptile, bird, mammal). This analysis yields highly interesting findings regarding conservation of
cell types across vertebrate evolution.

Major strengths of the paper are the unique spatial dataset that was collected, the careful
analysis of the spatial dataset and anatomical mapping, the extensive comparison with other
vertebrate telencephalon datasets to get a comprehensive evolutionary picture. Overall,
methods used are sharp, and careful statistical analyses has been performed. Quality tests for
snRNA-seq datasets were done comparably across the datasets used.
A weakness is the absence of comparison with other teleost models like zebrafish (see details
below). A final scheme including all species compared at adult stage would be informative. The
comparison with the mouse brain in Figure7e has issues that I have detailed below.

Major comments

An important aspect of the study is the comparison of their single-nuclei dataset with other
species datasets. Here the authors mention they used SAMap to calculate a similarity score that
was compared to permutations, and that they show those that have a score that is greater than
all permutations. This leads to different scales in figure 4 and 5 when comparing to mouse,
axolotl, turtle or songbird. Can they explain why these values are different? In addition, as each
scale is different, the reader cannot distill really to what extent the comparisons remain
meaningful. Is a mid-scale hit meaningful or not? Do the authors wish to say that darkest color
hits suggest true homology, and in the absence of that, a cell type might be "novel"? But since
we cannot really compare between species, the cutoff of a meaningful hit might be different on
different comparisons? It would be nice to better understand the view of the authors here. In
view of evolution, the comparison with all these species is a great resource!

Response: The reviewer's point is important, not all hits are equal, and interpreting hits should be easier
for the reader. We have added more detail and now focus on the strongest hits supported by the most
evidence. We have added visual cues to indicate reciprocal top hits, have included potential weakness
focusing on just cell-type or region analysis, and have added more statements in the text that inform the
reader that we report results that are consistent across analyses.

The scale bars are not directly comparable across comparisons. The value of the similarity score
is based on the mean number of k-nearest neighbors between species 1 and 2. Thus this value is
dependent on the number of cells in species 1 and 2. Additionally, the minimum and maximum of the
scale bars are set based on the comparison-specific distribution of similarity scores as to be pleasing to the
eye. To determine the meaningfulness of similarity scores, we performed permutation testing. Scores
greater than all permutations were reported as significant. Thus, the cutoff of a meaningful hit is indeed
different across comparisons. Our comparative approach is designed to identify populations that are
transcriptionally similar. To fully ascribe homology, other similarities, such as morphological and
electrophysiological similarities, must be found, but we contribute our transcriptional evidence to
proposed (or observed) homologies.

Cell-types may be significantly transcriptionally similar to multiple other cell-types and a darker
color does imply that the hit is stronger, but it does not necessarily mean that the darkest hit is truly
homologous while the others are not. It is possible that a cell-type could be homologous to multiple cell-
types in another species. Absence of significant hits for a cell-type does imply that there is not a
transcriptionally similar population in the dataset being compared, and could be due to that cell type not
being included in the tissue sample or indeed the cell-type might be "novel."

I missed the comparison with zebrafish, especially given the fact the authors suggest an
evolutionary comparative picture that goes against the current view on zebrafish telencephalon
organisation. Does the cichlid have a different telencephalic build-up or is the current zebrafish
map incorrect? What is known from other teleost brains like Medaka or goldfish? Including this
comparison would also give the authors the opportunity to discover novelty. At the moment, it
would be unclear if non-matching or less-matching cell types are cichlid specific or teleost
specific.

Response: We agree with the reviewer that a comparison to another teleost species will help reveal
whether our results are cichlid specific or even *Mchenga conophoros* specific. Thus, we compared our
cell-types to those from goldfish (a recently published atlas by Tibi et al 2023
<https://www.science.org/doi/10.1126/sciadv.adh7693>). The goldfish atlas was utilized because it is high-
quality, all cells came from the adult brain, and goldfish are frequently used in functional studies
including some to which we refer in this work. Analysis of the goldfish telencephalon revealed similar
finding to ours, including conservation of GABAergic subclasses, CA3-like populations, subiculum-like
populations, and cingulate/retrosplenial-like populations. Thus, we found strong conservation between
cichlid and goldfish cell-types in our direct comparative analysis of the teleost species. Furthermore, our
team has been part of an effort to sequence and analyze the clownfish telencephalon
(<https://doi.org/10.1101/2024.01.29.577753>), which also revealed strong correspondence between teleost
cell-types (cichlid and clownfish). Strong conservation within teleosts is perhaps not unexpected, since
we find correspondence with cell types in more divergent vertebrate groups. Detailed assessment of cell
type homology in fishes will require more species to be sampled using spatial approaches. We plan to
collect new data from other cichlid lineages and other teleost lineages to further address the question of
cell-type novelty in future papers.

We do not feel that our evolutionary picture goes against the current view on the zebrafish
telencephalon organization. Since the evolutionary picture of the zebrafish and teleost subpallium is well-
established and supported by our results, we assume the reviewer is referring to regions within the
pallium. For pallial regions, we present our understanding of an ongoing debate in the literature, mainly in
the Discussion and Supplemental Text. There is substantial evidence for the homology of the DI-v and the
hippocampus, but for other pallial regions, there is not an established consensus. Thus we feel that our
results provide valuable intellectual contributions to ongoing debates.

Perhaps, most controversial of our findings is that the DI-g more so resembles vertebrate dorsal
pallial derivatives rather than the medial pallium, which was proposed due to the transcriptional similarity
between this MC region and its corresponding cell-types to cortical populations (including the
retrosplenial cortex, cingulate cortex, and visual cortex). While some propose the DI-d/DI-g to be medial
pallial along with the DI-v, the partial pallial eversion theory first proposed by Wullimann and Mueller
(Teleostean and mammalian forebrains contrasted: Evidence from genes to behavior, 2004) -which was
proposed using a zebrafish model- is generally consistent with our findings. It is unclear as to which
current zebrafish map the reviewer is referring to, but as far as we are aware, there remains a lack of
agreement on the pallial arrangement in teleosts and the precise mammalian homologues of many of these
regions.

The scheme in Figure 7e compares “adult” mouse with cichlid telencephalon. Although the
mouse section was taken from a previously published paper, it depicts an embryonic mouse
brain, not an adult brain. The Allen brain atlas provides suitable coronal sections for comparison

of brain regions in the adult mouse. The issue with Figure7e is that here, the authors in fact
depict embryonic origin of regions MP, DP, LP, VP, subpallium and display it as if these were
fixed areas in the adult brain as well. The issue is that there is a lot of cell migration and cells
from VP and subpallium end up in other adult areas, making the adult picture more complex.
The question is to what extent the study, performed on adult specimen, can really disentangle
this complexity, given the limitations of resolution of the visium and despite the powerful
analysis the authors have done by implementing their single-nuclei dataset in the spatial data
(figure 3). It would be really helpful if the authors could clarify what assumptions they can or
want to make.

Response: Good catch, the previous depiction was of an embryonic mouse brain. Thus, we have adapted
an image of an adult brain instead (Cárdenas et al. 2020). In teleosts, the topography of homologous
structures to mouse regions is one the key ways to infer how the cichlid brain folds. Since this is the one
of the first spatially-resolved cellular atlases of the fish brain and given our comparative results, there is
an unique opportunity to contribute to ongoing debates about the fish brain. We acknowledge the
complexity of the adult brain and when comparing brain regions, we do not seek to make any further
assumptions than previously made by others (Porter et al 2020, Ganz et al 2016).

Minor comments

Reads were mapped to the Maylandia zebra reference genome- to what extent is this reference
genome similar to the Mchenga conophoros genome? Can the authors give numbers of
similarity on the transcriptomic sequence level?

Response: All Malawi cichlid species are quite closely related (e.g., a segregating site every ~500-1000
base pairs for genic sequence (Loh et al. 2008). It is customary to use this reference for comparisons
across the species flock (Malinsky et al. 2018; York et al. 2018; Johnson et al. 2023).

The authors identify cell types that seem to be GLUT and GABA: is this a unique teleost cell
type? can they explain?

Response: This is a good question. These cell-types are referred to as GABA/Glut because we observed
large numbers of nuclei that expressed glutamatergic neuronal markers as well as large numbers of nuclei
that expressed GABAergic markers within these clusters. Other clusters strongly tended to contain either
glutamatergic nuclei or GABAergic nuclei (as determined by marker genes). The name should not be
thought of as evidence for or against a unique cell-type in the teleost brain, but instead as the product of
our “primary” clustering parameters combined with our naming convention to help readers better track
the functional identities of clusters throughout the paper. For example, the “secondary” clusters within the
GABA/Glut primary cluster were all either glutamatergic or GABAergic. The naming convention for the
snRNA-seq cell-types is detailed in our previous paper (Johnson, Z. V. *et al.* Cellular profiling of a
recently-evolved social behavior in cichlid fishes. *Nat. Commun.* **14**, 1–19 (2023).), and for clarity to our
readers we have explicitly stated where to find the naming convention for these cell-types in our figure
legend for Figure 3B.

Line 200: wrong figure panel cited?

Response: Thank you for catching this. Fixed!

Lines 580-2: In contrast, our SAMap integration revealed no significant difference in the number
of

transcription factors (or other gene categories) driving relationships between cell-types in the
mouse neocortex and turtle aDVR/aDC.

The data sampled here are taken from adult fish, that might have already downregulated TF
expression. To what extent might this affect the authors' conclusions?

Response: Our data from fish does indeed come from adult subjects and the reviewer is correct that many
relevant transcription factors may have been downregulated in adulthood. In other comparative analyses
of scRNA-seq datasets from adult samples, transcription factors have been used as an indicator of
"regulatory networks that underlie conserved regional or cellular identity programs" (Colquitt et al 2020,
Tosches et al 2018). We agree with the implication of the reviewer that this may not be the most accurate
measure, especially in adult samples. Thus, we do not perform more similar analyses for other
comparisons nor do we stake our main points on transcription factor analysis. This analysis is mainly
performed in response to those that may expect it and we use it only in the same context it has been used
in the past (i.e. comparing sauropsid aDVR and aDC to the mammalian neocortex).

Lines 593-595 Taken together, our results and those from previous analyses[8,9] suggest that
conserved neuronal cell-types in the teleost forebrain may represent precursors to well-studied
neocortical populations in the mammalian brain.

The term "precursor" suggests these cell types might have the ability to mature or give rise to
mammalian neocortical cell types. In view of how evolution might have worked, this is rather
unlikely. Would it not be more likely that fish cell types x y z might be more similar to one or
more putative ancestral cell types that gave rise to an expanded array of cell types in the
mammalian neocortex? Can the authors rephrase this?

Response: Thank you for pointing this out, we have replaced our usage of the term 'precursors' in this
sentence to 'antecedents' to better illustrate our point.

This study is largely a meta-analysis using SAMap and no other tool was used to validate key
findings. The authors should address this in their discussion.

Response: Good point, this is now addressed in the discussion.

Figure 4A Zeisel et al-

Figure 4 similarity scores panel: axis cell types names are barely readable; this is an issue with
other figures as well; change size of those panels (if possible)?

Response: Thank you for pointing this out, we agree that the small size of these axes labels is not ideal.
Unfortunately, we are not able to make these cell-type labels any larger, and have made them as large as
we could.

REVIEWERS' COMMENTS:

Reviewer #1 (Remarks to the Author):

The authors have sufficiently addressed my previous questions. Highlighting reciprocal hits especially helps the reader better connect the claims with the results. I only have one minor question.

1- Can the authors clarify the subpallium and ventral pallium difference for cichlid fish in Figure 7e? The manuscript initially defines ventral pallium as subpallium ("The teleost telencephalon is subdivided into pallial (dorsal) and subpallial (ventral) domains") but they appear as different regions in Figure 7e.

Reviewer #2 (Remarks to the Author):

The changes the authors made have very much improved the manuscript, and I have no further comments.

Reviewer #3 (Remarks to the Author):

The authors comprehensively answered to all comments we raised. We are particularly happy with the addition of the comparison of their dataset to the goldfish dataset, which substantiates their findings. The figure legends are much clearer to the reader, and the elaboration of the discussion is much appreciated.

The authors might have missed it, but Figure 7e in the revised manuscript still contains the old mouse scheme instead of the one adapted from Cárdenas et al. 2020, as written in the legend (and would be a good solution indeed)... Can this still be fixed?

Reviewer #4 (Remarks to the Author):

The authors comprehensively answered to all comments we raised. We are particularly happy with the addition of the comparison of their dataset to the goldfish dataset, which substantiates their findings. The figure legends are much clearer to the reader, and the elaboration of the discussion is much appreciated.

The authors might have missed it, but Figure 7e in the revised manuscript still contains the old mouse scheme instead of the one adapted from Cárdenas et al. 2020, as written in the legend (and would be a good solution indeed). Can this still be fixed?

General Response: Below are the modified figures followed by detailed responses to the reviewer's
 comments.

 Figure 7. Panel e has been updated to include the mouse scheme from Cárdenas et al. 2020, as written in
 the figure legend.

Reviewer #1 (Remarks to the Author):

The authors have sufficiently addressed my previous questions. Highlighting reciprocal hits
 especially helps the reader better connect the claims with the results. I only have one minor
 question.

1- Can the authors clarify the subpallium and ventral pallium difference for cichlid fish in Figure
 7e? The manuscript initially defines ventral pallium as subpallium ("The teleost telencephalon is

subdivided into pallial (dorsal) and subpallial (ventral) domains”) but they appear as different
regions in Figure 7e.

The distinction is that the subpallium is the ventral domain of the *telencephalon* (as we stated, and as the
reviewer quoted), while the ventral pallium is the ventral domain of the *pallium* >> the pallium being the
dorsal domain of the telencephalon. We feel that this distinction is sufficiently clear in the text and
figures. However, the anatomical terminology in parentheses in the quote can be removed if you wish,
although we have included it by convention (see for instance our analysis of pallial and subpallial
developmental domains in the attached paper). Thank you for your detailed comments and time taken to
review our manuscript.

Attached paper: <https://www.nature.com/articles/ncomms2753>

Reviewer #2 (Remarks to the Author):

The changes the authors made have very much improved the manuscript, and I have no further
comments.

Thank you for all your comments, they were vital in improving the manuscript.

Reviewer #3 (Remarks to the Author):

The authors comprehensively answered to all comments we raised. We are particularly happy
with the addition of the comparison of their dataset to the goldfish dataset, which substantiates
their findings. The figure legends are much clearer to the reader, and the elaboration of the
discussion is much appreciated.

The authors might have missed it, but Figure 7e in the revised manuscript still contains the old
mouse scheme instead of the one adapted from Cárdenas et al. 2020, as written in the legend
(and would be a good solution indeed)... Can this still be fixed?

Great catch! Figure 7e has been updated to include the mouse scheme adapted from Cárdenas et al. 2020.
Thank you for your keen eye and comments.

Reviewer #4 (Remarks to the Author):

The authors comprehensively answered to all comments we raised. We are particularly happy
with the addition of the comparison of their dataset to the goldfish dataset, which substantiates
their findings. The figure legends are much clearer to the reader, and the elaboration of the
discussion is much appreciated.

The authors might have missed it, but Figure 7e in the revised manuscript still contains the old
mouse scheme instead of the one adapted from Cárdenas et al. 2020, as written in the legend
(and would be a good solution indeed). Can this still be fixed?

Great catch! Figure 7e has been updated to include the mouse scheme adapted from Cárdenas et al. 2020.
Thank you for your keen eye and comments.
